# Notch1 regulated autophagy controls survival and suppressor activity of activated murine T-regulatory cells

**Nimi Marcel[1,2], Apurva Sarin[1,3]***

[1]National Centre for Biological Sciences, Bengaluru, India; [2]Department of Biology, Manipal University, Manipal, India; [3]Institute for Stem Cell Biology & Regenerative Medicine, Bengaluru, India

**Abstract** Cell survival is one of several processes regulated by the Notch pathway in mammalian cells. Here we report functional outcomes of non-nuclear Notch signaling to activate autophagy, a conserved cellular response to nutrient stress, regulating survival in murine natural T-regulatory cells (Tregs), an immune subset controlling tolerance and inflammation. Induction of autophagy required ligand-dependent, Notch intracellular domain (NIC) activity, which controlled mitochondrial organization and survival of activated Tregs. Consistently, NIC immune-precipitated Beclin and Atg14, constituents of the autophagy initiation complex. Further, ectopic expression of an effector of autophagy (Atg3) or recombinant NIC tagged to a nuclear export signal (NIC-NES), restored autophagy and suppressor function in *Notch1*[-/-] Tregs. Furthermore, *Notch1* deficiency in the Treg lineage resulted in immune hyperactivity, implicating Notch activity in Treg homeostasis. Notch1 integration with autophagy, revealed in these experiments, holds implications for Notch regulated cell-fate decisions governing differentiation.

*For correspondence: sarina@ncbs.res.in

**Competing interests:** The authors declare that no competing interests exist.

## Introduction

Notch signaling is a conserved pathway, directed by ligand-dependent processing, culminating in the release of the Notch intracellular domain (NIC) from its membrane-tethered precursor, allowing its translocation to the nucleus to initiate transcription (*Kopan and Ilagan, 2009*). Signaling through Notch1 is obligatory for acquisition of T-cell fate (*Radtke et al., 1999*). However, Notch signaling is differentially activated during T-cell development and in mature T-cells, including the differentiation of induced Tregs (iTregs) from Foxp3[-ve] naive T-cell precursors (*Radtke et al., 1999*; *Laky et al., 2015*; *Maekawa et al., 2015*; *Yvon et al., 2003*; *Wolfer et al., 2001*). The transcription factor Foxp3 specifies Treg identity and unlike in induced (i)Tregs where Notch1 regulates Foxp3 expression in response to activation cues, in naturally arising Tregs, Foxp3 expression is developmentally regulated (*Fontenot et al., 2003*). Interestingly, activated Tregs are also distinguished by non-nuclear Notch1 signaling (*Perumalsamy et al., 2012*).

Macro-autophagy (autophagy) is a conserved catabolic process, which buffers cells from limiting nutrient conditions (*Nakatogawa et al., 2009*),and is controlled by the evolutionarily conserved Atg/ATG family of proteins (*Nakatogawa et al., 2009*; *Mizushima et al., 2011*). Autophagy is implicated in the regulation of diverse aspects of immunity including the differentiation and homeostasis of CD8[+] memory T-cells and Treg function amongst other cell types (*Deretic et al., 2013*; *Parekh et al., 2013*; *Xu et al., 2014*). The mechanism[s] by which autophagy is regulated in T-cell subsets are likely diverse and not characterized. In this study we describe Notch1 signaling to autophagy in the context of activated Treg survival and consequences to Treg function.

**eLife digest** A cell must have access to adequate amounts of nutrients if it is to survive and perform its role in the body. Some cells, including immune cells, must travel around the body to sites where nutrients may be in short supply. For example, a type of immune cell called the T-regulatory cell travels to wounds and other hostile places where other types of immune cells are actively fighting infections and/or damaged cells.

The T-regulatory cell's job is to reduce excess or harmful immune activity to prevent the immune system from damaging healthy cells in the body. However, it was not clear how they survive and protect themselves in these tough situations. In 2012, researchers showed that a protein called the Notch receptor helps the T-regulatory cells survive in conditions where nutrients are scarce.

Now, Marcel et al. – including some of the researchers involved in the 2012 study – have investigated how the Notch receptor protects mouse T-regulatory cells. The experiments show that T-regulatory cells grown in nutrient-poor conditions in the laboratory activate a protective mechanism known as autophagy. This mechanism recycles damaged or nonessential structures within the cell into nutrients. The Notch receptor is needed to trigger autophagy at the right time and the loss of this receptor causes the cells to die when they are starved of nutrients.

Further experiments found that the Notch receptor is essential for the T-regulatory cells to work properly in mice. Animals born lacking this receptor in T-regulatory cells quickly develop inflammation in response to even minor irritations. More studies are needed to reveal the details of this protective strategy, and to find out if it is used by other cells in the body.

Through the analysis of Tregs, activated by T-cell receptor (TCR) cross-linking in vitro, we show that (non-nuclear) Notch1 regulates survival via the activation of an autophagy-signaling cascade. In agreement with their central role in coordinating decisions of cell death and survival, mitochondria were a prominent cellular target, responsive to perturbations of the Notch-autophagy cascade. We validate these interactions by reconstituting the Notch-autophagy cascade in Notch1 deficient Tregs. We also show that Notch1 confers cytokine-independence via the activation of autophagy in T-effectors, which are derived from naïve T-cells. The consequence of Notch activity in Treg physiology is assessed in two contexts: One, the suppression of antigen-induced T-cell proliferation and correction of defective glucose clearance in genetically obese mice using adoptive transfers of test Treg populations. Second, the characterization of inflammation and increased activation of T-cell subsets occurring in mice with *Notch1* ablated in the Foxp3 (Treg) lineage. Collectively, these findings suggest a hitherto unappreciated role for the (non-nuclear) Notch-autophagy axis in the regulation of natural Treg function.

## Results

T-cells depend on cytokines for nutrient uptake and survival (*Purushothaman and Sarin, 2009*; *Vella et al., 1998*). A role for Notch1 activity in regulating survival in the absence of exogenous cytokines of activated Tregs has been reported (*Perumalsamy et al., 2012*), however, the cellular mechanisms activated by Notch1 in this context are not known. Autophagy, a conserved catabolic process is implicated in survival in response to nutrient stressors (*Lum et al., 2005*). Hence, we investigated if Notch signaling to autophagy regulates activated Treg survival following cytokine withdrawal. Unless otherwise mentioned, the analysis in this study is based on experiments with natural Tregs activated in vitro.

### Cytokine withdrawal triggers autophagy in Tregs

To assess a role for autophagy in Treg survival, activated Tregs are switched to complete medium, which contains serum but is not supplemented with the cytokine IL-2. Cells are monitored at various time points for induction of autophagy or survival following modulations described in the sections that follow. The recruitment of the microtubule-associated protein LC3 and its smaller lipidated form LC3II, into the autophagosome membrane is a molecular signature and necessary event in the progression of autophagy (*Kabeya et al., 2000*). The change in LC3 can be detected in immunoblots of

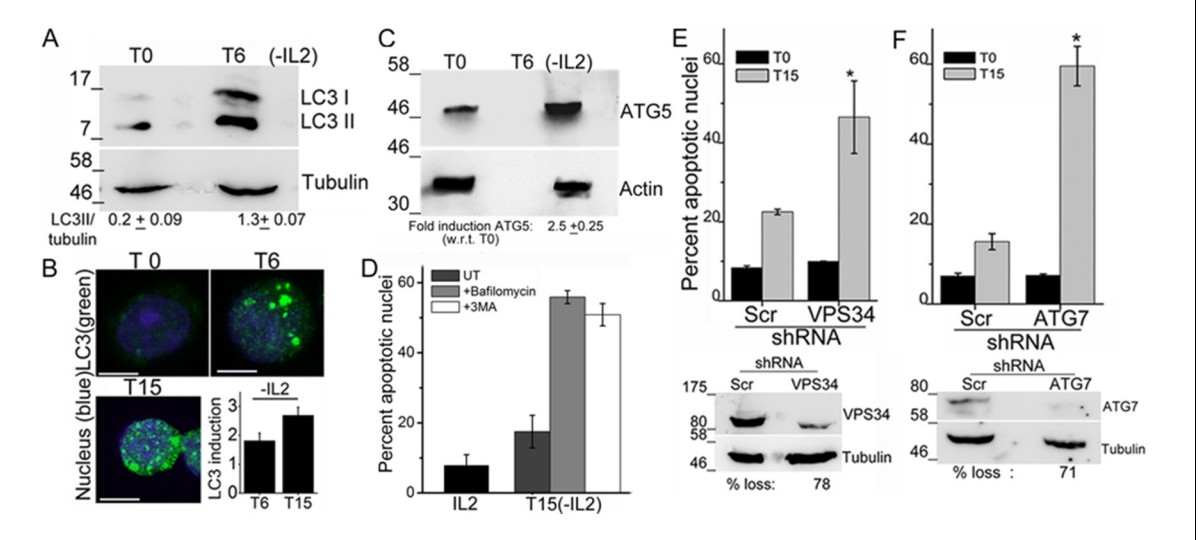

**Figure 1.** Autophagy is activated on cytokine withdrawal in activated Tregs. (**A**) Immunoblots probed for LC3 in lysates of Tregs at onset (T0) and after 6 hr culture without IL-2. The values below are densitometry analysis of LC3II relative to tubulin. (**B**) Z-projected confocal images of Tregs at onset (T0) and cultured without IL-2 for times indicated and stained for LC3 (green) and Hoechst 33342 (blue). Change in fluorescence intensities for LC3 relative to T0 are plotted. (n=150 cells/time point). (**C**) Immunoblot probed for ATG5 in lysates of Tregs cultured as described in **A**. (**D**−**F**) Apoptotic damage following 15 hr of IL-2 withdrawal in Tregs cultured in the presence of Bafilomycin (Baf) or 3-MA (**D**) or transduced with shRNA specific for VPS34 (**E**) or ATG7 (**F**) or a scrambled control (Scr). Immunoblots of scrambled and shRNA transfected cells are shown below. Data shown are the mean ± SD from at least 3 independent experiments, *p<0.03. Scale bar 5 μm. This figure is accompanied by *Figure 1—figure supplement 1*.

The following figure supplement is available for figure 1:

**Figure supplement 1.** Tregs activate autophagy in response to cytokine deprivation.

cell lysates, where the modified isoform is detected at a reduced molecular weight or by immunostaining intact cells when large puncta are marked by antibodies to LC3.

An increase in the LC3II isoform was detected in lysates of Tregs, which had been cultured without IL-2 for 6 hr, relative to the onset of the assay (T0) (*Figure 1A*). Immunostaining with the same antibody as used for the immunoblots and visualization of intact cells by confocal microscopy, showed that diffuse LC3 staining observed in Tregs at T0, progressively changed to large, readily visualized puncta by 6 hr, persisting till 15 hr following cytokine-withdrawal (*Figure 1B*). Quantifiable changes in fluorescence intensity and size of puncta were detected over this period (*Figure 1B* and *Figure 1—figure supplement 1A*). It should be noted that Tregs are viable throughout the course of this assay (*Figure 1—figure supplement 1B*). The protein Atg5, a molecular indicator of the activation of autophagy (*Mizushima, et al., 2011*), was also increased following cytokine withdrawal as detected by Immunoblots of Tregs cultured without cytokine (*Figure 1C*).

To assess if the activation of autophagy was necessary for Treg survival, inhibitors that block induction or progression of autophagy were tested. The inclusion of Bafilomycin A (Baf), which blocks autophagosome-lysosome fusion or 3-Methyladenine (3MA), at the onset of the cytokine deprivation assay resulted in cell death, when measured at 15 hr, indicating that Treg survival was abrogated (*Figure 1D*). To validate observations made with chemical inhibitors, we expressed shRNA to Vps34 (the lipid kinase regulating initiation of autophagy) or to Atg7, proteins implicated in autophagy progression, by retroviral infection coupled with antibiotic selection (as described in methods) to generate activated Tregs populations ablated for Vps34 or Atg7 protein expression. The loss of either VpS34 or Atg7 in activated Tregs resulted in cell death following cytokine withdrawal (*Figure 1E,F*), establishing a requirement for these intermediates in Treg survival.

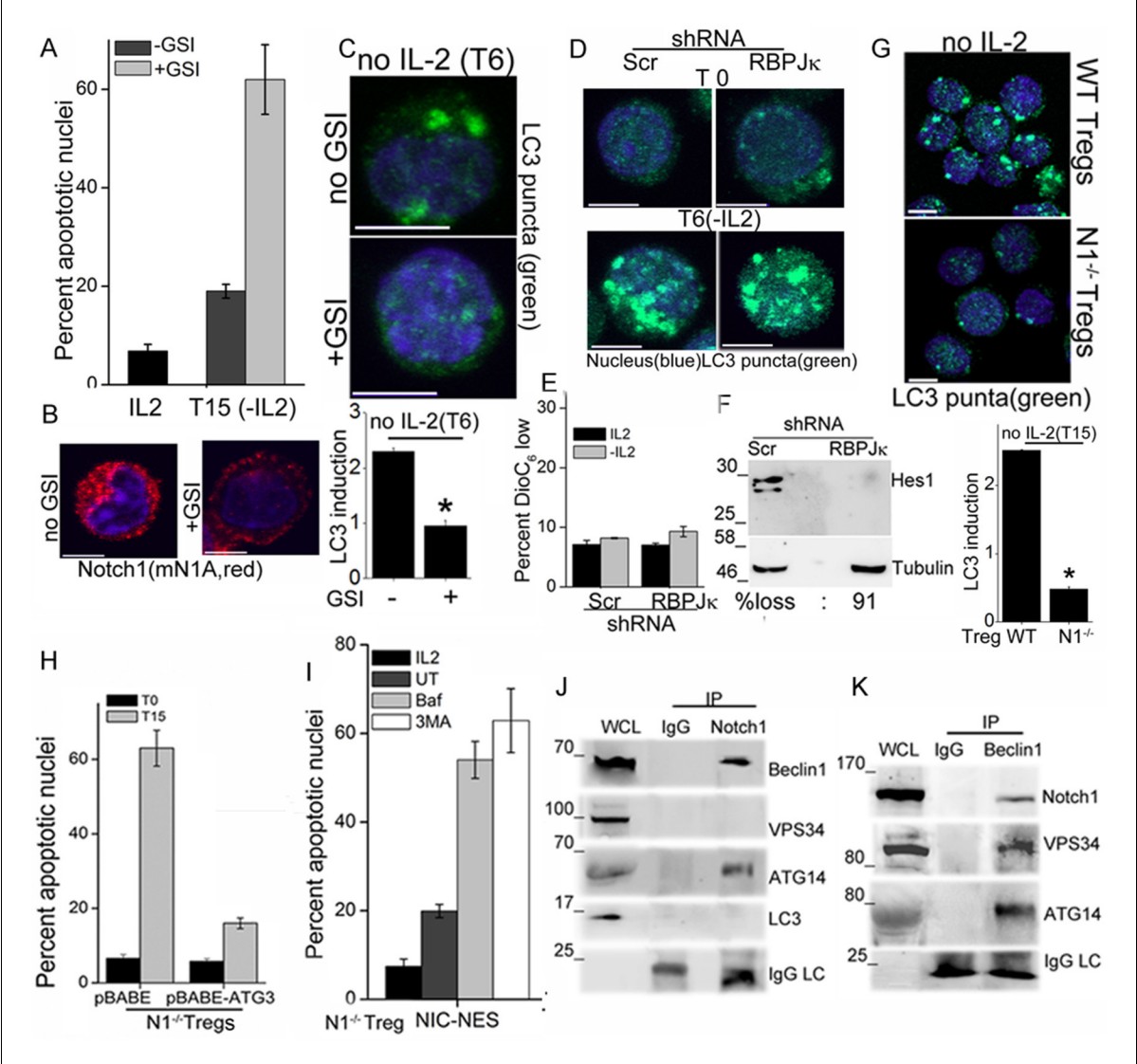

**Figure 2.** Notch regulates induction of autophagy in Tregs. (A) Apoptotic damage in Tregs cultured for 15 hr without IL-2, with or without GSI. (B) Representative images of Tregs cultured without IL-2, with or without GSI-X for 6 hr, stained for NIC (mNIA, red) and Hoechst 33342 (blue). (C) Representative, Z-projected images of Tregs treated as in (B) for 6 hr and stained for LC3 and Hoechst 33342. Plot below indicates the change in fluorescence intensities of LC3 relative to T0 (mean +/- SD, n=120 cell/ condition). (D) Tregs transduced with shRNA to RBPJκ or scrambled control, cultured without and with IL-2 for 6 hr and stained for LC3 and Hoechst 33342. (E) DiOC6 uptake in cells transduced with shRNA to RBPJκ or scrambled control and cultured with and without IL-2 for 6 hr. (F) Immunoblot analysis for Hes1 and tubulin in cells expressing control or RBPJκ shRNA. (G) Z-projected confocal images of wildtype (WT) or *Notch1⁻/⁻* Tregs, cultured without IL-2 for 15 hr and stained for LC3 and Hoechst 33342. Plot below indicates the change in fluorescence intensities of LC3 relative to T0 (mean +/- SD, n=120 cell/ condition). (H) Apoptotic damage measured in *Notch1⁻/⁻* Tregs transduced with recombinant ATG3 or pBABE vector control, at onset (T0) or after culture without IL-2 for 15 hr. (I) *Notch1⁻/⁻* Tregs transduced with NIC-NES cultured for 15 hr in the conditions indicated and scored for apoptotic damage. (J) and (K) Immune complexes precipitated from Treg lysates using indicated (IP) antibodies. Immunoblots were probed for proteins indicated on the right. Data show the mean ± SD of at least 3 independent experiments. In all micrographs, LC3 staining is in green and Hoechst 33342 in blue. Scale bar: 5 μm; *p≤0.03 This figure is accompanied by *Figure 2—figure supplement 1*.

The following figure supplement is available for figure 2:

**Figure supplement 1.** Notch1 signaling from the cytoplasm is needed for autophagy induction in Tregs.

## Notch1 activity regulates autophagy in Tregs

We next asked if Notch1 is required for activation of autophagy in activated Tregs in response to cytokine withdrawal. These experiments tested for the involvement of non-nuclear, ligand-dependent Notch1 activity, shown earlier to promote Treg survival (*Perumalsamy et al., 2012*). A pharmacological inhibitor of the enzyme γ-secretase, (GS inhibitor)-X, inhibits cleavage (S3 cleavage post ligand binding) and release of NIC. Culturing Tregs with GSI-X abrogated survival following cytokine withdrawal (*Figure 2A*). In GSI treated cells, immunostaining for Notch1 showed that that staining was restricted to the cell membrane, confirming that receptor cleavage is disrupted (*Figure 2B* and *Figure 2—figure supplement 1A*). Further, induction of LC3 puncta was not observed in Tregs cultured with GSI during cytokine withdrawal (*Figure 2C*, and *Figure 2—figure supplement 1B*), indicating that autophagy is not activated. Earlier work from the laboratory had identified a non-redundant role for the ligand Delta Like Ligand (DLL)-1 in Treg survival. Confirming ligand-dependence, ablation of DLL-1 in Tregs, compromised survival and induction of LC3 puncta in cells cultured without cytokine (*Figure 2—figure supplement 1C and D*). Contrastingly, but in agreement with published analysis of NIC1 signaling in Tregs, both the induction of LC3 puncta or cytokine-independent survival in Tregs was unchanged following shRNA-mediated ablation of RBPJ-κ, a co-factor of NIC transcription (*Figure 2D,E* and *Figure 2—figure supplement 1E*). The ablation of RBPJ-κ attenuated expression of Hes1 a target of NIC (*Figure 2F*) confirming efficacy of shRNA ablation. Taken together, these data suggested that ligand dependent Notch1 activity was required for autophagy and regulation was independent of canonical Notch interactions, such as those requiring RBPJ-κ (*Kopan and Ilagan, 2009*).

More direct evidence that Notch1 regulates activation of autophagy in response to cytokine withdrawal came from the analysis of activated *Notch1*[-/-] Tregs generated from mice with a targeted deletion of *Notch1* in the mature T-cell compartment (*Cd4-Cre::Notch1*[lox/lox] mice). The induction of LC3 puncta in response to cytokine withdrawal was blunted in *Notch1*[-/-] Tregs (*Figure 2G* and *Figure 2—figure supplement 1F*), and cell survival compromised in cells cultured without cytokine (*Figure 1—figure supplement 1B*). In consonance with dependence on autophagy, retroviral transduction of recombinant Atg3 in *Notch1*[-/-] Tregs restored survival following cytokine withdrawal (*Figure 2H*). Atg3 is an E2-ubiquitin-like-conjugating enzyme, which catalyzes the lipidation of the effector protein LC3I (ATG8) (*Mizushma et al., 1998*), a key step in autophagy progression. Atg3 is also implicated in maintaining mitochondrial health under conditions of nutrient stress (*Besteiro et al., 2011*; *Radoshevich and Debnath, 2011*). From these experiments we concluded that Notch1 activation of autophagy was necessary for Treg survival following cytokine withdrawal in culture.

NIC is enriched in the cytoplasm of Tregs (shown later and *Perumalsamy et al., 2012*), and we tested if non-nuclear NIC activity regulated autophagy following cytokine withdrawal. Towards this aim, we employed an approach used by others and us to achieve nuclear export of recombinant NIC (*Shin et al., 2006*). Thus, *Notch1*[-/-] Tregs were retrovirally transfected with recombinant NIC tagged to a Nuclear Export Signal (NIC-NES), which prevents nuclear residence. NIC-NES transfected Tregs express a recombinant predominantly localized in the cytoplasm (*Figure 2—figure supplement 1G*). NIC-NES expression protected *Notch1*[-/-] Tregs from apoptosis following cytokine-withdrawal (*Figure 2I*) and this protection is attenuated by the inclusion of Baf or 3MA (*Figure 2I*). Thus, non-nuclear NIC activity could activate autophagy signaling in Tregs following cytokine withdrawal.

We next assessed if proteins in the autophagy cascade immune-precipitated with NIC. In Tregs initiated into cytokine-deprivation, an antibody specific for Notch1 (mN1A) but not an IgG (isotype control) immune-precipitated Beclin1 and Atg14 but not Vps34 and LC3 (*Figure 2J*), indicating that NIC formed complexes with specific components of the autophagy pathway. In the reverse analysis, Beclin1 immune-precipitated NIC confirming the specificity of its association with NIC and this complex also included Atg14 and Vps34 (*Figure 2K*). Thus, Beclin is detected in complexes that include either Vps34 or NIC with both complexes including Atg14 as expected. Vps34 and Beclin are known to form multiple and dynamic cellular complexes, which are regulated by the phosphorylation of the proteins themselves as well their interacting partners (*Kang et al., 2011*; *Kim et al., 2012*). Thus the exclusion of Vps34 from the NIC complex may indicates interactions with proteins (such as Beclin) whose functions may be regulated by Notch activity. The data do not rule out transient associations between NIC and Vps34. That NIC formed complexes with proteins regulating autophagy in Tregs,

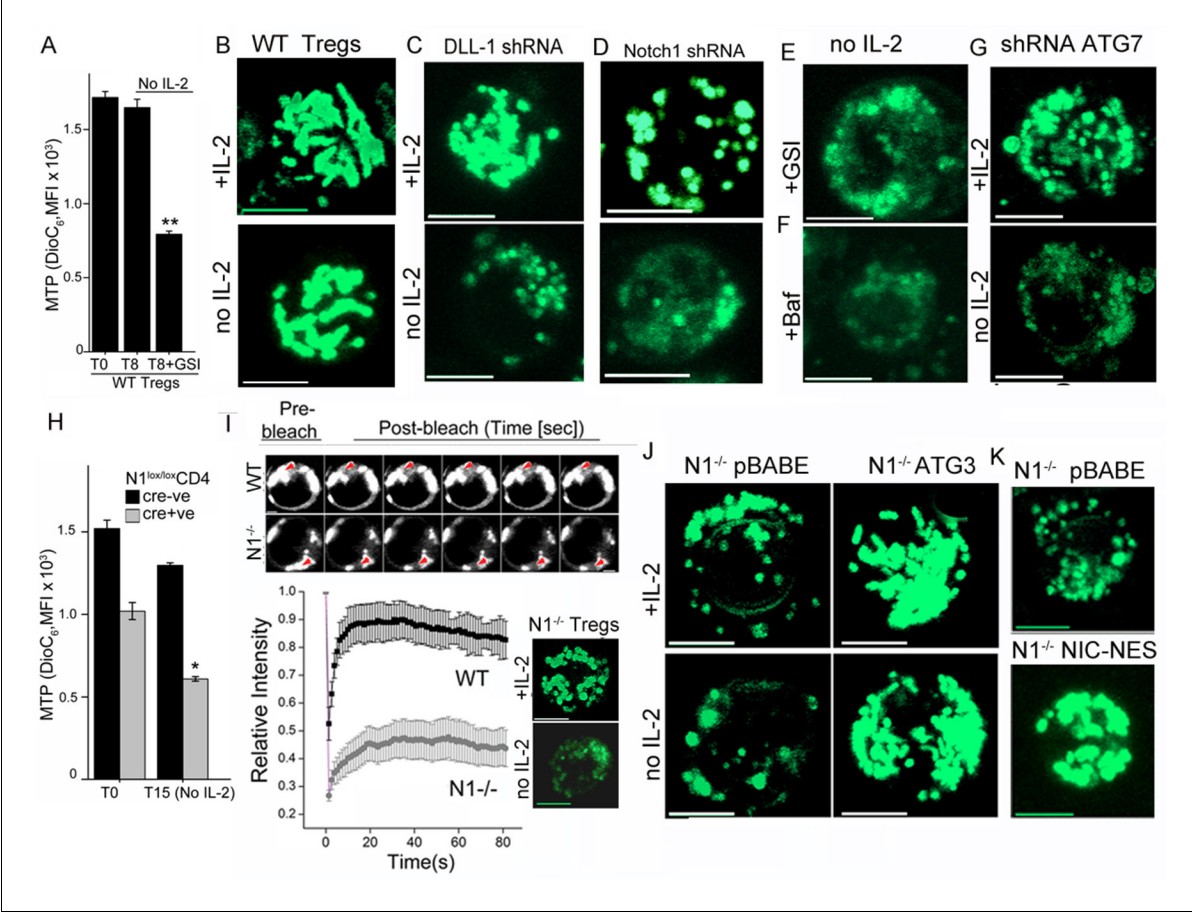

**Figure 3.** Notch regulates mitochondrial organization via autophagy. (A) DiOC$_6$ uptake in activated Tregs in the input population at onset of the assay (T0) or in cells cultured without cytokine for 8 hr with or without 10 µM GSI (Data are plotted to show Mean ± SD, p**</=0.001). (B-G) Representative confocal (Z-projected) images of mitochondria stained with MitoTracker Green in WT Tregs cultured for 6 hr in the following conditions: with or without IL-2 (B), post retroviral transfection of DLL-1 shRNA +/- IL-2 (C); or transfection of Notch1 shRNA +/- IL-2 (D); no IL-2+GSI (E) no IL-2 + Baf (F) or post retroviral transfection of shRNA to Atg7 +/- IL-2 (G). F and G, percent DiOC$_6$-high (live) WT and *Notch1*$^{-/-}$ Tregs at T0 (F) or 15 hr after culture without IL-2 (G). Mean ± SD from 3 separate experiments. (H) DiOC$_6$ fluorescence in activated *Notch1*$^{-/-}$ and control *Notch1*$^{+/+}$ Tregs in the input population at the onset of the assay (T0) or in cells cultured without cytokine for 15 hr. (Mean ± SD, p*</=0.03). (I) FRAP analysis in MitoTracker Green loaded WT or *Notch1*$^{-/-}$ Tregs (n=10 cells/ cell type, scale bar 2 µm) at T0. Inset: mitochondria in *Notch1*$^{-/-}$ Tregs cultured for 6 hr with or without IL-2. (J and K) Representative confocal images (at 6 hr) of mitochondria loaded with MitoTracker Green in *Notch1*$^{-/-}$ Tregs transfected with Atg3 or empty vector (pBABE), cultured with or without IL-2 (J) or *Notch1*$^{-/-}$ Tregs transfected with NIC-NES or empty vector (pBABE) in IL-2 (K). Images are representative of n=20 cells per experimental group from 2–3 experiments, scale bar 5 µm. This figure is accompanied by *Figure 3—figure supplement 1*

The following figure supplement is available for figure 3:

**Figure supplement 1.** Notch1 regulation of mitochondrial organisation is mediated via autophagy.

indicated a more direct role for Notch1 in the autophagic cascade, although the molecular regulation and dynamics of this interaction remains to be dissected.

Mitochondrial re-modeling and activity control cellular responses to changing bioenergetic needs, which facilitate transitions between nutrient-replete and deficient conditions (*Rambold et al., 2011*, *Hailey et al., 2010*). Since Notch modulation of mitochondrial function has been suggested in earlier work (*Perumalsamy et al., 2010*; *Kasahara et al., 2013*), mitochondrial organization and integrity in Tregs were next examined.

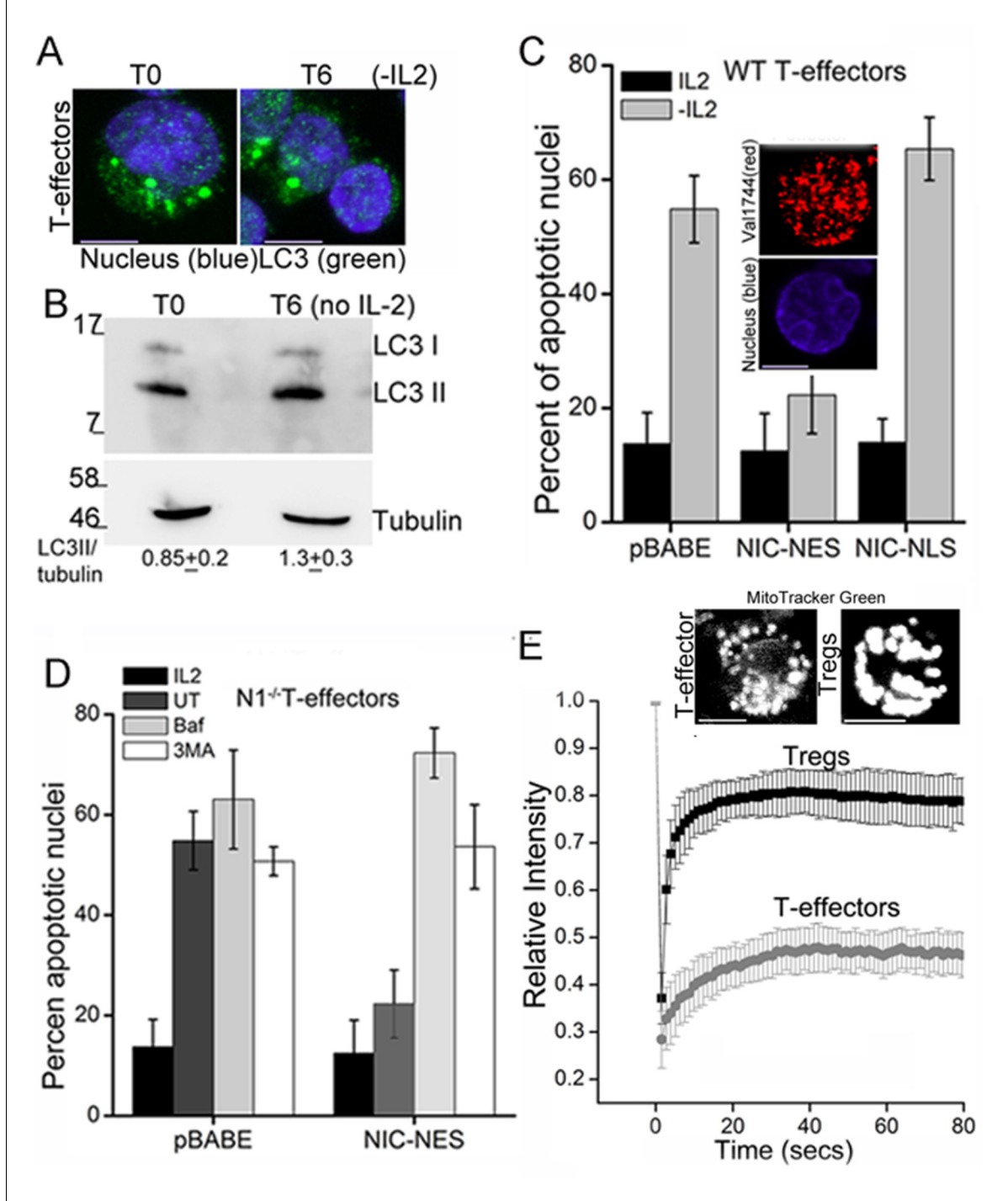

**Figure 4.** NIC-NES rescues cell death via autophagy in T-effectors. (A) Z-projected confocal images of T-effectors stained for LC3 (green) and Hoechst 33342 (blue) at onset of assay (T0) or following culture without IL-2 for 6 hr. (B) Immunoblots probed for LC3 in lysates of T-effectors at T0 and cultured without IL-2 for 6 hr. The values indicate densitometry analysis of LC3II relative to tubulin. (C) Apoptotic damage induced by IL-2 withdrawal in T-effectors expressing NIC-NES, NIC-NLS or pBABE cultured in the presence or absence of IL-2. Inset: Representative image of a T-effector stained with Val1744 antibody to detect endogenous Notch (red) and counterstained with Hoechst 33342 (blue). (D) Apoptotic damage in *Notch1*[-/-] T-effectors expressing NIC-NES or pBABE cultured with IL-2 (black), without IL-2 (grey) and the addition of Baf (light grey) or 3-MA (white) for 15 hr. (E) FRAP analysis in Tregs and T-effectors loaded with MitoTracker Green at T0 (n=10 cells/ cell type, scale bar 2 μm). Inset above, representative images of MitoTracker Green loaded cells visualized by confocal microscopy. Data shown are the mean ± SD from 3 independent experiments. Scale bar 5 μm. This figure is accompanied by *Figure 4—figure supplement 1*

*Figure 4 continued*

The following figure supplement is available for figure 4:

**Figure supplement 1.** Non-nuclear Notch1 dependent survival is mediated by autophagy in T-effectors.

## Mitochondrial integrity is dependent on NIC and autophagy

Mitochondria were analyzed employing a combination of biophysical and imaging methods in live cells and a potentiometric fluorescent probe, $DiOC_6$, which measures mitochondrial trans-membrane potential (MTP), an indicator of mitochondrial energetic state.

Uptake of the dye $DiOC_6$ is a well-established flow-cytometry based measure of MTP in intact cells. Activated Tregs undergo almost no change in MTP following cytokine withdrawal, consistent with their survival (*Figure 3A*). However, inclusion of GSI compromised mitochondrial function as it triggered a loss of MTP within 8–9 hr of culture (*Figure 3A*). Thus, mitochondrial activity was regulated by Notch in Tregs. We next asked if Notch activity also controlled mitochondrial organization in Tregs. In live activated Tregs, mitochondria marked with MitoTracker Green and visualized by confocal microscopy appear as interconnected structures, which remained so following cytokine withdrawal (*Figure 3B* and *Figure 3—figure supplement 1A*). Mitochondrial organization in cells cultured without cytokine was disrupted by perturbations of Notch activity, which included shRNA-mediated ablation of the Notch ligand DLL-1 (*Figure 3C* and *Figure 3—figure supplement 1B*) or shRNA mediated ablation of Notch1 (*Figure 3D*) or by the inclusion of GSI-X (*Figure 3E* and *Figure 3—figure supplement 1Cii*). Mitochondrial organization was also disrupted by the addition of Baf (*Figure 3F* and *Figure 3—figure supplement 1Ciii*) or by shRNA-mediated ablation of Atg7 (*Figure 3G*, and *Figure 3—figure supplement 1C iv,v*), implicating autophagic signaling in organelle integrity.

Uptake of $DiOC_6$ was significantly lower in $Notch1^{-/-}$ Tregs relative to WT Tregs cultured in IL-2 (*Figure 3H*). Following cytokine withdrawal, further loss of MTP, indicated compromised mitochondrial function in $Notch1^{-/-}$ Tregs (*Figure 3H*). We compared mitochondrial contiguity in WT and $Notch1^{-/-}$ Tregs using the technique of fluorescence recovery after photo bleaching (FRAP) in live cells loaded with MitoTracker Green. Fluorescence recovery in WT Tregs was high, which is consistent with a connected morphology of the organelle (*Figure 3I*). Low FRAP in $Notch1^{-/-}$ Tregs is indicative of discontinuous structures (*Figure 3I*), which was confirmed by microscopy (*Figure 3I* inset and *Figure 3—figure supplement 1D*). The punctate morphology of mitochondria in $Notch1^{-/-}$ Tregs was reduced, with the organelle appearing more tubular and connected in cells expressing recombinant Atg3 (*Figure 3J* and *Figure 3—figure supplement 1E*) or recombinant NIC-NES (*Figure 3K* and *Figure 3—figure supplement 1F*) relative to cells expressing an empty vector control (*Figure 3J,K* and *Figure 3—figure supplement 1F*). Thus a loss of MTP and mitochondrial fragmentation was associated with poor survival outcomes following cytokine withdrawal in Tregs. Our experiments established that NIC and autophagy signaling were critical for the integrity and function of mitochondria in these cells.

## NIC activates autophagy to promote survival of T-effectors following cytokine withdrawal

We next assessed if Notch signaling to autophagy was a more generalized mechanism that can be activated in other cells. For this, we employed T-effectors generated from naïve T-cell precursors, as T-effectors do not survive cytokine withdrawal (*Purushothaman and Sarin, 2009*) and we had prior evidence that ectopic expression of recombinant NIC protected T-effectors from apoptosis in this context (*Bheeshmachar et al., 2006*).

T-effectors - generated as described in methods by TCR stimulation of naïve T-cells - do not increase LC3 puncta or the LC3II isoform following cytokine deprivation (*Figure 4A,B* and *Figure 4—figure supplement 1A*). Endogenous NIC is nuclear localized in T-effectors unlike Tregs (*Figure 4—figure supplement 1B*). As was seen with Tregs, the ectopic expression of NIC-NES protected T-effectors from apoptosis triggered by cytokine withdrawal (*Figure 4C*). However a recombinant NIC modified by the inclusion of a Nuclear Localization Signal (NIC-NLS, *Figure 4C* inset), which enforces localization to the nucleus did not confer protection from cell death (*Figure 4C*). We ruled out a role

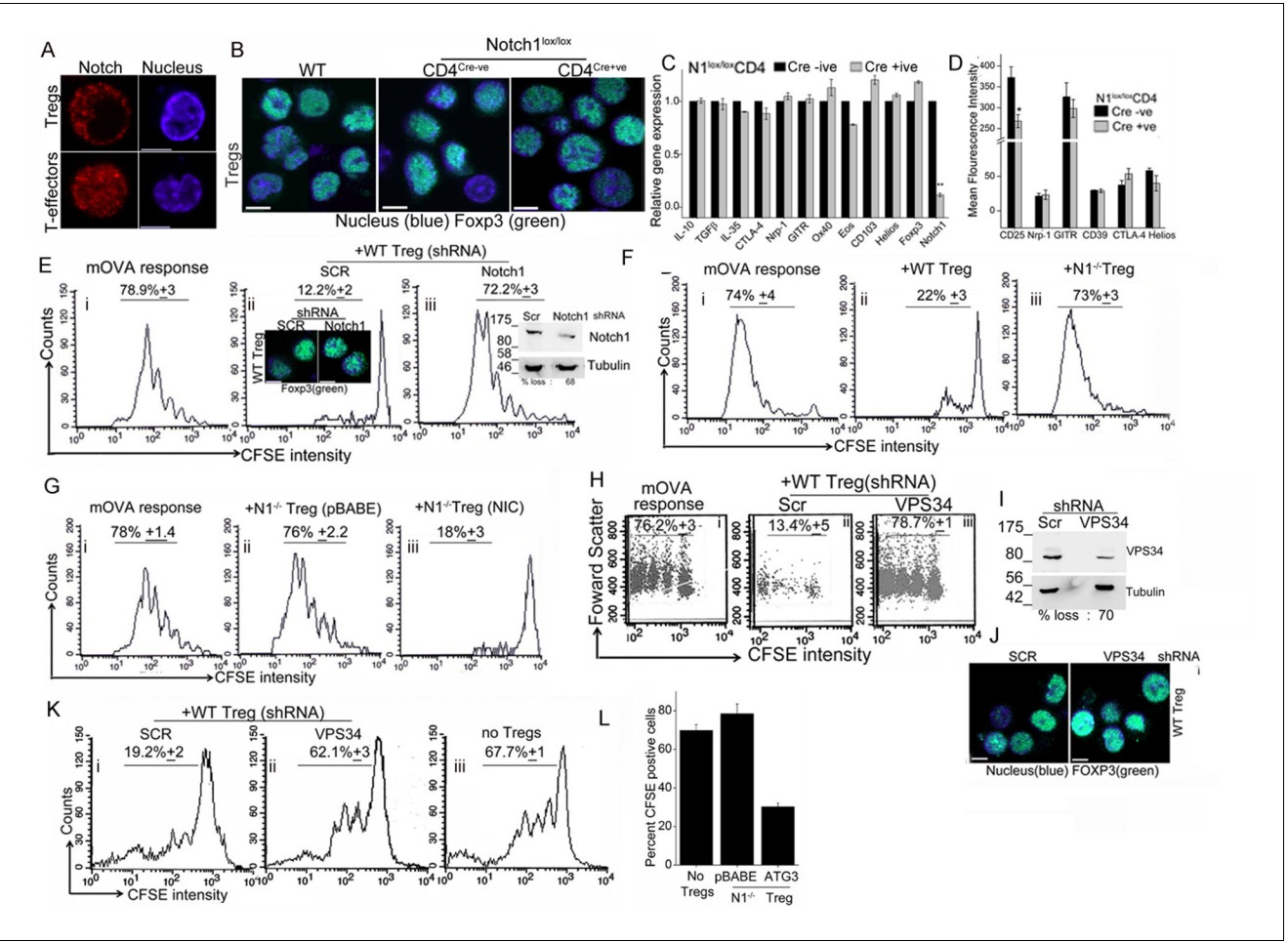

**Figure 5.** Notch activity and autophagy regulate Treg suppressor function. (**A**) Representative, confocal images (central stack) of activated Tregs (upper panel), or T-effectors (lower panel) immune-stained for NIC (red) and Hoechst 33342 (blue). N: 15–20 cells/experiment. (**B**) Representative confocal (central stack) field views of activated Tregs immune-stained for Foxp3 (green) and Hoechst 33342 (blue). Tregs were derived from C57Bl/6 (wildtype, WT) or *Cre* negative (Cre-ve) or *Cd4-Cre::Notch1^{lox/lox}* (Cre+ve) mice. (**C**) Real Time PCR quantification of genes enriched in Tregs, comparing activated Tregs from *Cd4-Cre::Notch1^{lox/lox}* (open bars) and genetic control *Cre*-ive (black bars) mice. 5–6 mice are included in each group being compared. The data plotted is mean+/-SD. p**</=0.001. (**D**) Flowcytometry based expression of molecules (mean fluorescence intensity, MFI, relative to control isotype antibody shown) enriched in Treg subsets, compared in activated Tregs generated from Cre+ive (open bars) and Cre-ive (black bars) mice. 4–6 mice are included in each group. (**E**) Flowcytometry plots indicating dilution of CFSE in CD45.2^{+} (OT-II) gated cells isolated from lymph nodes of mice injected with OT-II cells alone (i) or, OT-II co-injected with Tregs transduced with scrambled (ii) or Notch1 shRNA (iii), three days after antigen challenge. Inset: (ii) confocal images of Tregs detecting Foxp3 in scrambled or Notch1 shRNA groups and (iii) immunoblot for Notch1 in shRNA treated groups. (**F**) Flowcytometry plots indicating dilution of CFSE in CD45.2^{+} (OT-II) gated cells isolated from lymph nodes of mice injected with OT-II (i) OT-II + WT Tregs (ii) or OT-II + *Notch1^{-/-}* Tegs (iii) three days after antigen challenge. (**G**) CFSE dilutions of OT-II cells co-injected with *Notch1^{-/-}* Tregs transduced with empty vector (pBABE) (ii) or recombinant NIC (iii) three days after antigen challenge. Data are representative of 2–3 independent experiments with 2–3 mice/ experimental group. Percentage of cells in the CFSE diluted group is indicated in each plot. Tregs activated *in vitro* are used in all experimetns. (**H**) Proliferation in CD4^{+}OT-II cells alone (i), or co-injected with Tregs transduced with retroviruses expressing shRNA to VPS34 (iii) or a scrambled control (ii) post antigen challenge. (**I**) Confocal (merged) images of Foxp3 (green) immunostaining counterstained with Hoechst 33342 (blue). (**J**) immunoblot detecting VPS34 (H) in shRNA treated groups as in H. (**K**) Flow cytometry plots indicating CFSE dilution in naïve CD4+T-cells 72 hr post-stimulation with anti-CD3 and APC in vitro. T-cells were either cultured alone (no Tregs) or with Tregs transduced with shRNA as described in H. (**L**) Percent CFSE positive, CD45.2+ OT-II cells isolated from host mice isolated after antigen challenge. Host mice were injected with OT-II naïve T-cells alone (no Tregs) or, naïve cells co-injected with *Notch1^{-/-}* Tregs retrovirally transduced with empty vector pBABE or recombinant ATG3. Scale bar: 5 µm. This figure is accompanied by *Figure 5—figure supplement 1*.

The following figure supplement is available for figure 5:

**Figure supplement 1.** Notch1 signaling and autophagy pathway are needed for Treg function.

for endogenous Notch1 in this context, by reproducing the protective effect of NIC-NES in T-effectors derived from *Cd4-Cre::Notch1*$^{lox/lox}$ naïve T-cells (*Figure 4D*). Further, protection from cell death was abrogated if inhibitors of autophagy - Baf or 3MA - were included in culture (*Figure 4D*). Notably, mitochondria in T-effectors do not demonstrate contiguity in FRAP assays (*Figure 4E* and *Figure 4—figure supplement 1C*) or by confocal image analysis (*Figure 4E*). Thus, the analysis indicated that the crosstalk between Notch1 and autophagy for the regulation of survival is not restricted to Tregs alone.

Survival is necessary for cellular function and an important component of homeostasis. The experiments described thus far had revealed a role for Notch signaling to autophagy for activated Treg survival in culture. We next tested if the NIC-autophagy signaling axis was also of consequence to Treg functions in vivo.

## Perturbations of Notch1 or autophagy inhibit suppressor activity in activated Tregs

The transcription factor Foxp3, specifies Treg identity and suppressor function and while it can be induced in naïve T-cells by appropriate cytokines (*Samon et al., 2008*; *Mota et al., 2014*), Foxp3 expression is developmentally regulated in naturally arising Tregs (*Fontenot et al., 2003*). As reported earlier (*Perumalsamy et al., 2012*), NIC is enriched in the cytoplasm of activated Tregs (*Figure 5A*, upper panel and *Figure 4—figure supplement 1B*), which contrasts with the more expected pattern of nuclear localization in T-effectors (*Figure 5A*, lower panel and *Figure 4—figure supplement 1B*). We noted a small but consistent elevation in the number of Tregs recovered from *Cd4-Cre::Notch1*$^{lox/lox}$ (2.25 ± 0.3) relative to Cre negative genetic controls (1.8 ± 0.3) mice per 100 million spleen cells. However, Foxp3 expression was comparable in activated Tregs generated from *Cd4-Cre::Notch1*$^{lox/lox}$ mice and the matched genetic controls, including Tregs from C57Bl/6 (wild-type) mice (*Figure 5B* and *Figure 5—figure supplement 1A*, lower panel). We recapitulated Notch1-dependence in activation-induced expression of Foxp3 in induced Tregs. Thus, in culture conditions that polarize to the generation of iTregs, in contrast to the genetic control (Cre negative) littermates, naïve *Cd4-Cre::Notch1*$^{lox/lox}$ T-cells, cannot be differentiated to express Foxp3, (*Figure 5—figure supplement 1A* upper panel).

Before we assessed functional capabilities of Tregs of the two genotypes, we compared gene and protein expression of a subset of molecules associated with Treg activation and function. Activated Tregs generated from *Cd4-Cre::Notch1*$^{lox/lox}$ mice and their genetic (Cre negative) controls were compared for molecules enriched in Treg subsets. Gene-expression analysis by microarray was confirmed using RT-PCR analysis and in some instances flow-cytometry analysis for proteins. In this analysis, activated Tregs of the two genotypes showed no striking differences, with loss of Notch1 transcript serving as the positive (internal) control (*Figure 5C* and *Figure 5—figure supplement 1B*). Similarly, comparison of gene sets enriched for cytokines associated with Tregs revealed no differences in the *Cd4-Cre::Notch1*$^{lox/lox}$ or genetic (Cre negative) control mice (*Figure 5—figure supplement 1C*). Similar trends were observed in flow-cytometry based analysis of cell surface and intracellular markers implicated in Treg differentiation and function (*Figure 5D* and *Figure 5—figure supplement 1D*). These experiments indicated a small difference in the expression of the gene *Eos* and high-affinity IL-2 receptor expression. A better understanding of the functional significance of these changes awaits analysis of other gene groups using combination of approaches described here.

The distribution of naïve and memory T-cell subsets in circulation, an indication of the resting vs. activated state of the immune system was comparable in Tregs derived from *Cre* negative controls and *Cd4-Cre::Notch1*$^{lox/lox}$ mice (*Figure 5—figure supplement 1E*). Based on the analysis of transcripts, protein expression and immune subsets we expected no difference between wildtype and *Notch1*$^{-/-}$ Tregs in functional assays of suppression. Indeed, in earlier work when tested for suppressor activity in vitro, activated Tregs of the two genotypes were comparable (*Perumalsamy et al 2012*). However, suppression is a multi-step process with distinct demands that operate on cells in vivo, hence we tested Tregs from *Notch1*$^{-/-}$ and genetic controls in assays of immune-suppression in vivo.

Established protocols (*Quah et al, 2007*; *Klein et al., 2003*), involving the transfer of responder and suppressor cells into host (immune-competent) mice were followed to measure the ability of Tregs to suppress antigen-induced T-cell proliferation. In this assay, purified naïve (OT-II) T-cells

loaded with a fluorescent dye – chosen because it partitions into daughter cells following division - were injected into host mice, which are subsequently challenged with ovalbumin as OT-II cells respond to this antigen. The OT-II T-cells can be distinguished from host cells by antibody-mediated detection of the CD45.2 molecule on their cell surface (*Figure 5—figure supplement 1F*). Three days post antigen challenge lymph node cells are isolated from the host, cells are gated on CD45.2$^+$ and the CFSE profile scored in this group using two-color flow-cytometry. The distribution of CFSE loaded cells spans multiple dilutions of lower intensity, indicative of proliferation (*Figure 5Ei*), whereas in mice not challenged with antigen, CFSE will be detected at the highest intensity (*Figure 5—figure supplement 1G*, filled histogram). Since functional Tregs suppress T-cell proliferation, co-injection of Tregs and CFSE loaded naïve T-cells, results in reduced/no dilution of CFSE (*Figure 5Eii*) following antigen challenge. Thus, the CFSE dilution in antigen-responsive T-cells in a co-injection (adoptive transfer) protocol indicates the percentage of T-cells proliferating in the absence or presence of Tregs. Additionally, by loading Tregs with CFSE and tracking transferred OT-II T-cells by the expression of CD45.2, it can be demonstrated that both the subsets localize to the same lymph node, thereby enabling this analysis (*Figure 5—figure supplement 1H*).

Following the principle described above, naïve T-cells isolated from OT-II (CD45.2$^+$) mice, were loaded with the fluorescent dye CFSE as described in methods. CFSE loaded cells were injected into congenic hosts, (CD45.1$^+$, B6SJL mice), which differ only in the expression of the CD45 isoform. Naïve T-cells were co-injected without or with in vitro activated (CD45.1$^+$) Tregs, which were earlier transfected with either shRNA to Notch1 or a scrambled control. Host mice were challenged with the appropriate antigen (mOVA), 15–18 hr after receiving cells. Three days post challenge, cells were isolated from draining lymph nodes of host mice and analyzed for CFSE dilution in the gated CD45.2$^+$ subset donor OT-II T-cells (*Figure 5E*, panel i). In mice co-injected with Tregs transfected with scrambled shRNA, proliferation of OT-II cells was expectedly blunted and CFSE fluorescence in OT-II cells detected at the highest intensity of CFSE indicating an undivided population (*Figure 5E* panel ii). On the other hand, in cells isolated from mice injected with Tregs transduced with Notch1 shRNA, OT-II T-cells proliferated robustly (*Figure 5E* panel iii), and were comparable to proliferation in OT-II cells injected without Tregs (*Figure 5E*, panel i). This indicates that Notch1 controls the suppressor activity of Tregs. The expression of Foxp3 was not changed by Notch1 ablation (Figure 5Eii, inset and *Figure 5—figure supplement 1I*). Similarly, suppressor activity was attenuated in Notch1$^{-/-}$ Tregs (*Figure 5F*, compare panels, i and iii) as compared to wildtype Tregs (Tregs$^{WT}$) (*Figure 5F* panel ii). However, reconstitution with recombinant NIC restored suppressor activity of Notch1$^{-/-}$ Tregs (*Figure 5G*, compare iii and ii).

While the data suggest that Notch1 activity can regulate Treg suppressor function, these results are in contrast to an earlier observation from the laboratory wherein Notch1$^{-/-}$ Tregs showed activity comparable to control Tregs in vitro co-culture suppressor assays (*Perumalsamy et al., 2012*), which we can reproduce (data not shown). Thus, the suppressor assay in the culture dish appears to recapitulate a subset of the diverse cues present in vivo. Another possibility is that the accumulation of IL-2 produced by T-effectors in the co-culture assay, allows for the survival and hence suppressor activity of Notch1$^{-/-}$ Tregs in vitro. The in vivo assay was a more sensitive read-out of Notch activity in Tregs and we next tested if modulating autophagy in activated Tregs modified suppressor activity.

In agreement with the reported requirement of autophagy in Treg function (*Parekh et al., 2013*, *Wei et al., 2016*), shRNA to Vps34 abrogated Tregs$^{WT}$ suppressor activity relative to the cells treated with the scrambled shRNA (*Figure 5H* compare iii with ii) as compared to the control group injected without Tregs (*Figure 5Hi*). The ablation of Vps34 (inhibition of autophagy) did not modulate the expression of Foxp3 (*Figure 5I,J*). However, in an in vitro assay of Treg function, which measures suppression of conventional T-cell proliferation via stimulation of T-cell receptor in the presence of antigen presenting cells, the ablation of VpS34 in activated Tregs abrogated suppressor activity (*Figure 5K*, compare i and ii). Responder T-cell proliferation was comparable to proliferation of cells stimulated without Tregs (*Figure 5K*, iii). Thus, Vps34 was critical for Treg function and its requirement revealed in suppressor assays performed in vitro or in vivo. Furthermore, suppressor activity in assays in vivo was restored in Notch1$^{-/-}$ Tregs transfected with recombinant Atg3 (*Figure 5L* and *Figure 5—figure supplement 1J*). Expression of Atg3 had also protected cells from apoptosis, shown in earlier experiments.

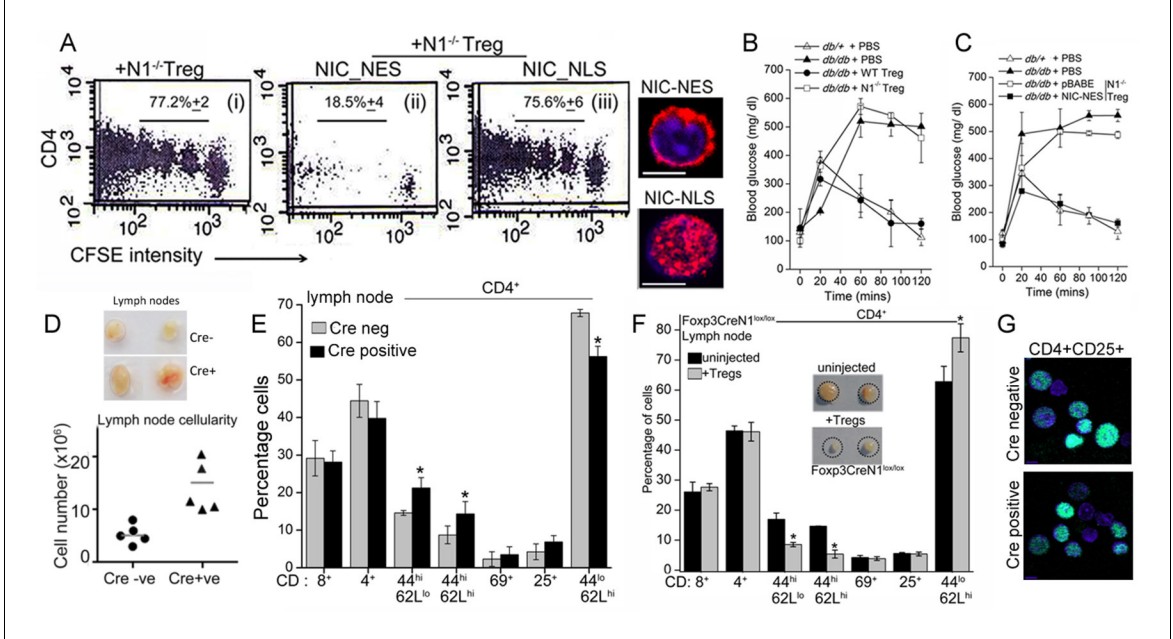

**Figure 6.** Non-nuclear NIC regulates Treg suppressor activity. (**A**) Proliferation of CFSE loaded CD4[+] OT-II cells co-injected with untransfected, or NIC-NES or NIC-NLS transfected *Notch1*[-/-] Tregs. The numbers indicate the percentage of the population with CFSE dilution Inset: staining for Notch1 in transfected cells. (**B**) IPGTT in *Lepr*[db/+] (△) or *Lepr*[db/db] (filled △) mice injected intravenously with PBS (△ or▲), or WT (●) or *Notch1*[-/-] (□) Tregs. (**C**) IPGTT in *Lepr*[db/+] (△) or *Lepr*[db/db] mice injected intravenously with *Notch1*[-/-] Tregs expressing recombinant NIC-NES (■) or pBABE (□). Data are mean ± SD of 2 independent experiments with three animals in each condition. (**D**) Whole lymph nodes (representative) from Cre+ and Cre- mice and plotted below, cell recoveries from lymph nodes from mice from different experiments. (**E**) T-cell immune cells subset analysis in lymph nodes isolated from wild type and mutant mice. (**F**) Subset analysis of T-cells in lymph-nodes of mutant mice, which are either not injected or injected with WT Tregs mice seven days prior to analysis. (**G**) Representative confocal images (field views) of activated Tregs from *Cd4-Cre::Notch1*[lox/lox] and *Cre*[-ve] mice fixed and stained for Foxp3 (green) and counterstained with Hoechst 33342 (nucleus, blue). *p≤0.03; Scale bar: 5 µm This figure is accompanied by *Figure 6— figure supplement 1*.

The following figure supplement is available for figure 6:

**Figure supplement 1.** Treg suppressor function is regulated by Notch1 signaling from the cytoplasm.

## Non-nuclear Notch1 activity is a positive regulator of Treg function

Since Notch1 modulated Treg function, we tested if this was mediated through non-nuclear Notch activity. In the OT-II T-cell proliferation assay, expression of recombinant NIC-NES restored suppressor activity in transferred *Notch1*[-/-] Tregs (**Figure 6Aii**). Suppressor activity was not restored in *Notch1*[-/-] Tregs transfected with NIC-NLS (**Figure 6Aiii**). All comparisons were made with *Notch1*[-/-] Tregs transfected with a vector control group (**Figure 6Ai**), in which condition suppressor activity is not detected.

Next we tested the requirement of non-nuclear Notch and Atg3 activity in Treg function, in another model of immune-inflammation. The assay tracks the clearance of glucose injected into the blood stream of obese mice, and builds on the observation that impaired glucose clearance (also referred to as glucose intolerance) is a feature of obesity (**Winer et al., 2009**). In this context, the injection of Tregs has been shown to correct defects in the clearance of glucose, although protection is short-term and persists for 10–15 days (**Cipolletta et al., 2012**, **Feuerer et al.,2009**). The underlying mechanism of correction by Tregs is not understood, as there are no deficiencies in Treg number or function reported in obese (leptin receptor-deficient, *Lepr*[db/db]) mice (**De Rosa et al., 2007**). Nonetheless, the correction of defective glucose clearance remains a reproducible assay of Treg function.

In agreement with published observations we show that the clearance of (injected) glucose from blood in fasting *Lepr*[db/db] mice, is substantially delayed compared to heterozygous (*Lepr*[db/+]) littermates (**Figure 6B**). However, in *Lepr*[db/db] mice injected with Tregs[WT], and tested 7–8 days later,

clearance of blood glucose is comparable to *Lepr*[db/+]animals (*Figure 6B*). Correction is transient and is lost within 2 weeks following adoptive transfer (not shown). These data recapitulate published observations made with Tregs in this model (*Eller et al., 2011*). Hence, we next tested *Notch1*[-/-] Tregs in this assay system. Activated *Notch1*[-/-] Tregs transfected with the control vector did not improve glucose clearance, however, transferring NIC-NES expressing *Notch1*[-/-] Tregs, restored the clearance of glucose to rates comparable to *Lepr*[db/+] mice (*Figure 6C*), indicating that enrichment of non-nuclear Notch activity restored *Notch1*[-/-] Tregs activity in this assay. Similarly, as seen in the T-cell proliferation assay, injecting *Notch1*[-/-] Tregs transfected with recombinant Atg3, also corrected the rate of glucose clearance to a level comparable to that of control, non-obese mice (*Figure 6—figure supplement 1A*). Collectively, the experiments confirmed an important role for Notch1 activity in Treg function.

Another line of evidence implicating Notch1 in Treg homeostasis came from mice with a targeted deletion of *Notch1* in the Treg lineage (i.e. under the control of the *Foxp3* promoter). Mice with a *Notch1* ablation, were born in reduced numbers (1:16 as against the expected 1:8 ratio), presented with features consistent with dysregulated immune function and some similarities with *Scurfy* mice, which lack Tregs because of a loss-of-function mutation in the *Foxp3* gene (*Fontenot et al., 2003*). *Foxp3-Cre::Notch1*[lox/lox] mice (analyzed from 4–8 weeks of age) were viable, presented with enlarged lymph nodes and increased cellularity relative to *Cre*-negative littermate controls (*Figure 6D*). Increased cellularity resulted from increased numbers of CD4[+]CD62L[low]CD44[high] (effector memory) and CD4[+]CD62L[high]CD44[high] (central memory) T-cell subsets (*Figure 6E*). Changes in the CD4[+]CD25[high] and CD4[+]CD69[+] subsets, which are expressed on Tregs and are early markers of T-cell activation respectively, were inconsistent, showing small increases in number in some mice or remaining comparable to genetic controls in others (*Figure 6E*). No striking changes were observed in B-cell and myeloid subsets (*Figure 6—figure supplement 1B*). Further, in *Foxp3-Cre::Notch1*[lox/lox] mice, which had been injected with WT Tregs, seven days prior to analysis, lymph node inflammation and the number of activated CD4[+] T-cell subsets in lymph nodes was substantially reduced (*Figure 6F* and *Figure 6—figure supplement 1C*). This suggested that defects in Treg function might contribute to the observed inflammation in *Foxp3-Cre::Notch1*[lox/lox] mice. In preliminary experiments, the adoptive transfer of *Notch1*[-/-] Tregs did not reduce lymphoid accumulation (not shown). Notably, several *Foxp3-Cre::Notch1*[lox/lox] mice were smaller in size than littermate controls, had shorter, roughened fur and developed crusting of skin around the eyes by 4–5 weeks (*Figure 6—figure supplement 1D,E*). Nonetheless, Foxp3 expression, assessed by immunostaining in Tregs was comparable in mice of the different genotypes (*Figure 6G*). The analysis of *Notch1*[-/-] Tregs in functional assays of inflammation together with the phenotypes of mice with a deletion of *Notch1* in Tregs suggests an important role for the receptor in immune homeostasis.

## Discussion

Tregs control inflammatory responses mounted by the immune system (*Buckner, 2010*; *Sakaguchi, 2004*). In this study we position Notch1 regulated autophagy as an integral controlling element of activated Treg homeostasis. This is inferred from the observations that survival and function are compromised following perturbations of Notch signaling or autophagy in Tregs[WT] and the restoration of autophagy and suppressor activity following reconstitution of *Notch1*[-/-] Tregs with the nuclear-excluded recombinant NIC-NES or Atg3. Of note, the requirement for Notch1 in thymus-derived activated Tregs is distinct from identified roles for Notch-TGFβ interactions in iTregs (*Samon et al., 2008*; *Mota et al., 2014*).

More generally, our experiments suggest that Notch-induced autophagy is a mechanism of quality control of mitochondrial architecture. This was also indicated by the restoration of organelle connectivity if NIC-NES or Atg3 were expressed in *Notch1*[-/-] Tregs or in T-effectors. The interactions between the organelle remodeling machinery observed earlier (*Perumalsamy et al., 2010)*) and the Notch-autophagy signaling axis, underpinning these outcomes remain to be characterized.

Cell survival in complex and changing environments associated with inflammation is an important component determining immune cell function. The induction of cell death in Tregs in response to cytokine deprivation appeared to correlate with suppressor function in vivo with one exception. The loss of suppressor activity in *Notch1*[-/-] Tregs in vivo, did not agree with outcomes of suppressor assays in vitro, wherein Notch1 appeared to be dispensable for Treg suppression

(*Perumalsamy et al., 2012*). Based on the understanding of Notch activity in Tregs we speculate that *Notch1-/-* Tregs suppress T-cell proliferation in vitro as concentrations of cytokines and growth factors that build up in the dish likely protect *Notch1-/-* Tregs from death. As Foxp3 levels are not compromised in these cells, this will allow suppressor activity. That the ablation of Vps34 compromised Treg activity in both in vivo and in vitro assays, is consistent with autophagy controlling Treg survival *and* playing a defining role in maintenance of functional identity (*Wei et al., 2016*), i.e. regulating Foxp3 expression. While we did not detect a loss in Foxp3 expression in the 24–48 hr period following ablation of Vps34, the eventual loss of Foxp3 expression and compromised survival as shown by Wei, et al., may underlie defective suppressor activity observed in the 3–4 day duration suppressor assays.

The dynamic regulation of Notch and its role as a critical determinant of Treg differentiation, function and homeostasis has emerged from work by another group (*Charbonnier et al.,2015*). Similarly, a more recent report (*Wei et al., 2016*), focused on events at the earlier stages of Treg undergoing activation to define a critical role for autophagy during differentiation of Tregs. Although focusing on different stages of (natural) Treg development, taken together, the studies suggest dynamic regulation of Notch activity (*Charbonnier et al.,2015*) or autophagy (*Wei et al., 2016*) in Treg homeostasis, which support the conclusions from our work albeit with some differences.

Unlike the study by Charbonnier et al., which positions RBPJ-k dependent Notch signaling as a negative regulator of Treg function we find that deletion of *Notch1* in the Treg lineage resulted in features of immune-inflammation consistent with a role for Notch1 in Treg function. However, both the studies implicate non-canonical Notch1 activity as necessary for maintenance of Treg identity. We speculate that the increased representation of recently activated and T-memory subsets in *Foxp3-Cre::Notch1lox/lox* mice, observed in our experiments, is likely a response to antigens experienced in the high barrier – but not SPF1 – conditions, that mice are housed in. Notably, genetic (littermate) controls were indistinguishable from wild type mice indicating the absence of overt infection. Interestingly, the phenotypes of autoimmunity reported in *Atg7* deficient mice (*Wei et al., 2016*), align with lymphoid proliferation and increased cellularity we observe in *Foxp3-Cre::Notch1lox/lox*mice. The interaction between Notch and autophagy revealed in our experiments suggests a hitherto unappreciated role for Notch signaling in the regulation of this process. Whether the effects of autophagy is executed through Notch1 signaling at all stages of Treg activation remains to be investigated.

Our experiments have focused on activated Tregs, an approach that identified a defining role for mTORC1 in Treg homeostasis (*Zheng et al., 2013*). Multiple studies have confirmed that the deficiency of Notch1 at later stages of T-cell development does not impair lineage commitment of Tregs. We propose however, that Notch1 tunes a late event in the differentiation of Tregs already committed to the suppressor lineage. This is consistent with our observation that low-levels of immune challenge elicit features of inflammation as compared to the *Cre* negative littermates.

Notch is implicated in instructive fate choices in the T-cell lineage, with commitment to T-cell fate the earliest amongst these (*Kopan and Ilagan, 2009*). Here we demonstrate a critical role for nonnuclear Notch1 activity, wherein Notch-autophagy interactions follow from immune stimulation and are important determinants of Treg function as evidenced in assays of adoptive transfers. Despite defects in *Notch1-/-*Treg function, mice with an ablation of Notch1 in mature CD4+T-cells did not present overt inflammatory phenotypes, which likely reflect its role in tuning responses of effector T-cells (*Laky et al., 2015*). However, the targeted deletion of *Notch1* in the Treg lineage resulted in inflammatory features suggested an integral role for Notch1 in Treg homeostasis. The possibility that Notch1 activity in Tregs is critical in specific contexts and redundant or non-essential in others cannot be excluded (*Zhou et al., 2015*). It is tempting to speculate that the integration of autophagy and non-nuclear Notch activity may be a conserved mechanism that tunes cell-fate decisions governed by the receptor in other cell types.

## Materials and methods

### Mice

The *Notch1lox/lox* and *Cd4-Cre::Notch1lox/lox* (*Notch1-/-*) strains were a gift from Freddy Radtke (*Wolfer et al., 2001*). *Foxp3tm4(YFP/Cre)*Ayr/J, C57BL/6J, B6SJL, OT-II and *Leprdb/db* strains were

obtained from the Jackson Laboratory. *Notch1*[lox/lox] and *Foxp3tm4(YFP/Cre)*Ayr/J strains were crossed to generate *Foxp3-Cre::Notch1*[lox/lox] mice. *Notch1*[lox/lox] mutant mouse stains were backcrossed with C57BL/6 mice. Except where specified, all experiments used mice within the age group of 8–12 weeks. Mice were housed in controlled temperature and light environments that are maintained in high barrier conditions with specific IVC (individually ventilated cages) controlled systems. The housing environment is tested and routinely monitored for the full pathogen panel recommended by FELESA (Federation of Laboratory Animal Science Associations). Breeding colonies were maintained in-house and all experimental protocols were approved by the Institutional Animal Ethics Committee (NCBS-AEC-AS-6/1/2012; INS-IAE-2016/01[N]) and are in compliance with the norms of the Committee for the Purpose of Control and Supervision of Experiments on Animals, Govt. of India.

## T-cells and retroviral transductions

To activate cells, CD4[+]CD25[+] natural Tregs were isolated from murine spleens (using a combination of negative selection to enrich for CD4[+] cells and a second step of positive selection for CD25[+] cells) following manufacturers instructions (R&D Systems or Invitrogen). The CD4[+]CD25[+] cells thus isolated were confirmed to be ~95% Foxp3[+] by immune-staining followed by confocal microscopy based analysis across multiple experiments. Cells were activated (~$2x10^6$/ml) by co-culturing with beads coated with antibodies to CD3 and CD28 (20ul/ ml; Invitrogen) in 24 well plates. After 48 hr, beads were removed by magnetic separation, and activated Tregs continued in culture in 50% conditioned medium and IL-2 (2 ng/ml) (R&D Systems) or used in assays. For the generation of induced Tregs, naïve CD4[+] T-cells isolated by negative selection as above were activated ($1x10^6$/ml) using beads coated with antibodies to CD3 and CD28 (20 ul/ml; Invitrogen) in presence of IL-2 (2 ng/ml) and TGF-β (2 ng/ml) for 72 hr. Generation of T-effectors and retroviral generation and infection of T-cells was as previously described (*Perumalsamy et al., 2012*). Briefly, retroviruses were packaged in HEK 293T using the packing vector pCLEco. The viral supernatant was concentrated and cells were infected after 24 hr stimulation by spinfection in 24 well plates (500 *g* for 90 min at 32°C) and continued in culture. After 48 hr, cells were harvested and continued in medium supplemented with IL-2 (1 ug/ml) for another 18–24 hr. Culture conditions included IL-2 (1 ug/ml) and the antibiotic Puromycin (1 ug/ml) for 48 hr to enrich for infected cells. Live cells were selected on day 2 by centrifugation in Ficoll, prior to use in functional assays. Knockdowns were assessed by Western blotting analysis of cell lysates (0.3–0.5 x$10^6$ cells per lysate) post antibiotic selection.

The plasmids pBABE, pBABE-NIC-NLS, pBABE-NIC-NES were gifts and have been described before (*Perumalsamy et al., 2010*). pBABE-puro-ATG3 was from Addgene (MA, USA). shRNA specific for VPS34, ATG7, RBPJ-k, Dll-1, Notch1 and scrambled control were from Origene.

## Apoptosis assays

Activated Tregs were cultured with or without IL-2 (0.3 x $10^6$/ml). After 15–18 hr, cells were harvested and tested for the induction of apoptotic damage. Nuclear morphology was scored in 200–300 cells across five fields in coded samples stained with Hoechst 33,342 (1 μg/ml) for 3–5 min at ambient temperature. Cells were stained with DiOC$_6$ (40nM in PBS) for 10 min at 37°C, washed to remove excess dye, re-suspended in PBS and mitochondrial transmembrane potential analyzed by flowcytometry.

## Western blot and immunoprecipitation analysis

Cell lysates were prepared using 0.4 x $10^6$ cells. Briefly, cell pellets were re-suspended (by vortexing) in 20–25 ul of SDS lysis buffer (2% SDS, glycerol, bromophenol blue, 1 M DTT and 1 M Tris-Cl pH 6.8 supplemented with a protease inhibitor cocktail - aprotinin, leupeptin and pepstatin (2 μg/ml each), 10 uM MG132, 1 mM PMSF, 1 mM NaF and 1 mM Na$_3$VO$_4$) and boiled for 10 min at 100°C. Whole cell lysates were resolved by SDS-PAGE and transferred to nitrocellulose membrane, (GE Healthcare) and incubated overnight at 4°C with primary antibodies at concentrations recommended by the manufacturers. The membrane was washed thrice with TBS-Tween20 followed by HRP-conjugated secondary antibody (CST, 1:1000 dilution) for 1 hr at ambient temperature. Membranes were developed using an ImageQuant LAS 4000 Biomolecular Imager (GE Healthcare) and quantified with Image J software.

For immune-precipitation analysis, $4 \times 10^6$ Tregs were lysed for 30 min at 4°C on a rotational cell mixer in 1%NP40 buffer (50 mM Tris, 1 mM NaCl, 1 mM EDTA) supplemented with aprotinin, leupeptin and pepstatin (2 μg/ml each), 10uM MG132, 1 mM PMSF, 1 mM NaF and 1 mM $Na_3VO_4$. Debris is removed by centrifugation and the supernatant incubated with primary antibody or IgG control (10 μg) for 1 hr at 4°C on a rotational cell mixer. The Immune complexes were precipitated for 2 hr at 4°C using washed Sepharose G plus beads (70 ul) on a rotational cell mixer. Beads bound to complexes were washed five times with ice-cold PBS by centrifugation at 1700 rpm. Finally, beads were boiled in SDS lysis buffer for 10 min before western blot analysis.

Primary antibodies used for western blot analysis include LC3 (D3U4C), Atg7 (D12B11), VPS34 (D9A5) Atg5 (D5F5U), Beclin-1 (D40C5) and Atg14 were from Cell Signaling Technology (used at a dilution of 1:500); NICD (clone mN1A) and DLL-1 (C-20) from Santa Cruz Biotechnology (used at adilution of 1:250); α-tubulin and actin (used at a dilution of 1:250) from Neomarker and Hes-1 (used at a dilution of 1:250) was from Millipore.

## Analysis of mitochondria

T-cells adhered to poly-D-lysine coated cover-slips were stained with 100 nM MitoTracker Green for 20 min at 37°C in complete medium, washed to remove excess dye and imaged with a Zeiss LSM Meta 510 as Z-stacks (1.0 μm, 3 zoom) using Plan-Apochromat 63 × NA 1.4 oil-immersion objective. Images were de-convoluted using Zeiss LSM software and stacks re-merged with Image J for Z-projections. In all experiments involving confocal microcopy based mitochondrial analysis; images of 20–25 cells per experimental conditions, across 2–3 independent were taken.

For FRAP analysis, cells were imaged using Zeiss LSM Meta 510 microscope (oil immersion objective, 63 ×. 0.9 NA). A confocal system with an integrated FRAP module, collected images every 2 s immediately after photo bleaching (circular ROIs of 1.5–2.0 μm diameter). Fluorescence recovery was analyzed after correcting for photo bleaching and background noise. FRAP measurement was performed on a minimum of 10 cells/ cell type from 3 independent experiments.

## Immune staining

T-cells ($2 \times 10^6$) adhered to poly-D-lysine–coated dishes (1 ug/ml in PBS coated for 15 min at RT) were fixed with 2% paraformaldehyde for 20 min at RT, permeabilized with neat methanol for 12 min on dry ice or at -20°C (LC3); 0.1% Tween-20 or 0.2% NP40 for 10 min at RT (Val1744 or Foxp3 and mNIA clones respectively) and then blocked with 5% BSA at RT. Samples were stained with primary antibodies overnight at 4°C (at dilutions of 1:100 for Val1744, 1:50 for mN1A, 1:100 for Foxp3) and secondary antibodies for 1 hr at room temperature. Antibodies were from the following sources: cleaved Notch1

(Notch-Val1744) and LC3 (D3U4C) were from Cell Signaling Technology; NICD (clone mN1A) from Santa Cruz Biotechnology and Foxp3 (FJK-16s) was from eBiosciences. Secondary antibodies were from Invitrogen and used at a dilution of 1:1000. Samples were imaged on Zeiss LSM Meta 510 as Z-stacks (1.0 μm, 3 zoom); Plan-Apochromat 63 × NA 1.4 oil-immersion objective. The stacks were processed to remove background based on secondary controls and Z-projected using Image J software. For all experiments involving imaging based analysis n = 20 cells/ condition in every experiment, across 3 independent experiments. For immunophenotyping, cells isolated from lymph nodes of mice with the required genotypes were stained with antibodies to indicated cell surface proteins and analyzed by flowcytometry. The analysis of cell surface markers was made on the lymphocyte population gated for size in the Forward scatter (Fsc) vs. Side scatter (Ssc) plot.

## Quantitative real-time PCR

RNA was isolated from $4 \times 10^6$ activated Tregs using a TRIzol (Invitrogen)-based RNA extraction protocol following the manufacture's instructions. cDNA was prepared using Superscript II (Invitrogen). Quantitative PCR was performed with SYBR Green (Thermo Scientific) in triplicate. Relative expression was calculated using the using the ΔΔ threshold cycle method ($2^{-\Delta\Delta CT}$). Primers sequences are as follows:

CTLA-4 Fwd:GGACGCAGATTTATGTCATTGATC,CTLA-4 Rev:CCAAGCTAACTGCGACAAGGA
TGF-Beta Fwd:CAACGCCATCTATGAGAAAACC, TGF-Beta Rev:AAGCCCTGTATTCCGTCTCC
GITR Fwd:AACGGAAGTGGCAACAACAC, GITR Rev:CTTGGGGCACAGAGGAAGA

IL-10 Fwd:GAAGACCCTCAGGATGCGG, IL-10 Rev:CCTGCTCCACTGCCTTGCT
IL-35 Fwd:CAATCACGCTACCTCCTCTTT,IL-35 Rev:AGTTTTTCTCTGGCCGTC
CD103Fwd:CGTGGAGAAGAAGGCAGAGT, CD103 Rev:TCGGGGGTAAAGGTCATAGAT
Eos Fwd:CCAAGTCCCTGAGTGGTTGT, Eos Rev:TTATCCAGGAGCCGTTCATC
Helios Fwd:ACACCTCAGGACCCATTCTG, Helios Rev:TCCATGCTGACATTCTGGAG
Neuropilin-1 Fwd:AGCAAGCGCAAGGCTAAGTC, Neuropilin-1Rev:ATCCTGATGAACCTTG
TGGAGAGA
Ox40 Fwd: CGAATTCCACCATGTATGTGTGGGTTCAG, Ox40Rev:CGGGATCCTCAGGAGCCAC-
CAAGGTGGG
Foxp3 Fwd:CACCTATGCCACCCTTATC, Foxp3 Rev:TCCTCTTCTTGCGAAACTC
HPRT Fwd:TCAGTCAACGGGGGACATAAA, HPRT Rev:GGGGCTGTACTGCTTAACCAG

Activated Tregs from CD4Cre$^{-ve}$ and CD4Cre$^{+ve}$ mice were also analyzed using Affymetrix Mouse_GXP_8X60K AMADID: 49,771 using Gene Spring GX Software.

## Suppressor assays and adoptive transfer protocols

CD4$^+$ naïve T cells were isolated from OTII transgenic mice spleens by negative selection using magnetic bead based separation protocols specified by manufacturers (MagCellect, R & D System). Freshly isolated native T-cells ($2 \times 10^6$ cells/ml in pre-warmed PBS) were loaded with 5uM CFSE and incubated for 8 min at 37°C (water bath). Cells were washed with complete medium to remove excess dye before use in suppressor assays. $0.5 \times 10^6$ CFSE loaded CD4$^+$ naïve OT-II T-cells, were co-injected with $0.3 \times 10^6$ activated Tregs intravenously (i.v) into congenic B6SJL host mice. 24 hr later, host mice were subcutaneously injected with 30 ug maleylated-ovalbumin (OVA) in CFA. 3 days later host lymph nodes were analyzed for CSFE dilution in the donor cells by gating on the CD45.2$^+$ population by flowcytometry. For the in vitro suppressor assay, $10 \times 10^4$ CFSE loaded naive CD4$^+$T-cells were activated in 96 well flat bottom plate (in triplicates) with soluble anti-CD3 (250 ng/well) and Mitomycin C (50 ug/ml) treated APC ($5 \times 10^4$) in the presence of activated Tregs (1:4::activated Treg: T-cells). 72 hr post-stimulation cells pooled from replicate wells were stained for CD4 expression and CFSE dilutions assessed in the CD4$^+$ cells by flowcytometry.

Rescue of lymphadenopathy in *Foxp3-Cre::Notch1$^{lox/lox}$* mice was based on a published protocol (*Fontenot et al., 2003*). Briefly, $1 \times 10^6$ freshly isolated WT Tregs were adoptively transferred (i.v.) into 4-week old *Foxp3-Cre::Notch1$^{lox/lox}$* mice. Seven days later, single cell suspensions were prepared from the lymph nodes (axillary and inguinal) of each of the mice injected with Tregs and matched un-injected controls. Lymph node cells were gated on the lymphocyte population based on size by the Forward scatter (Fsc) *vs.* Side scatter (Ssc) plot and analyzed for cell surface markers by flowcytometry using a BD FACSCalibur Cell Analyzer.

## Intra-peritoneal glucose tolerance test

Mice fasted for 6 hr were injected intra-peritoneally (ip) with 150 µl D-Glucose (2 g/kg body weight) in water. D-Glucose (Fischer scientific) - 2 grams per kilogram of body weight (g/kg) prepared in autoclaved water - was injected intra-peritoneally (i.p.) into *Lepr$^{db/db}$* and *Lepr$^{db/+}$* mice fasted for 6 hr. Before injecting the glucose solution the base-line blood glucose level was checked. After injecting the glucose solution, blood samples were obtained from the tail tip at the indicated times, and blood glucose concentrations were measured using a handheld glucometer (Contour TS, Blood glucose meter, Bayer). In experiments testing in vivo function of Tregs, $0.8 \times 10^6$ activated Tregs (*Notch1$^{-/-}$* or genetic controls) were adoptively transferred into the *Lepr$^{db/db}$* mice by the intravenous route using the tail vein. 7 days after adoptive transfer, IPGTT was performed on these mice.

## Data analysis

Data shown are the mean ± SD from 3 independent experiments, unless stated otherwise. Statistical significance was calculated with the two-population Student's *t* test. Data plots for FRAP assays are mean ± SEM from 3 experiments. Images from confocal microscopy were analyzed using Image J software (NIC, USA) and Adobe Photoshop. The confocal image stacks were processed to remove background based on secondary controls and Z-projected using Image J software. The Region of Interest (ROI) restricted integrated density was quantified for each cell in Image J software and normalized for area to calculate mean pixel intensity.

## Acknowledgements

We gratefully acknowledge Freddy Radtke (EPFL, Laussane) for generous gifts of the *Notch1*$^{lox/lox}$ and *Cd4-Cre::Notch1*$^{lox/lox}$ mice. We thank Dr. Chaitrali Saha who assisted with generation of cell preparations and manipulations involving mice. We acknowledge the Animal Care and Resource Centre (National Mouse Resource Grant) and the Central Imaging and Flowcytometry Facility (CIFF) at the National Centre for Biological Sciences (NCBS) and Institute for Stem Cell Biology and Regenerative Medicine (inStem), Bangalore; Sunil Laxman (inStem, Bangalore) and L Perumalsamy (IITM, Chennai) for comments on the manuscript; funding (BT/PR13446/COE/34/30/2015) from the Department of Biotechnology (DBT), India and NCBS and inStem to AS and a Senior Research Fellowship (09/860[0137]/2012-EMR-1) from the Council of Scientific and Industrial Research (CSIR), India to NM.

## Additional information

### Funding

| Funder | Grant reference number | Author |
|---|---|---|
| Department of Biotechnology , Ministry of Science and Technology | BT/PR13446/COE/34/30/2015 | Apurva Sarin |
| Council of Scientific and Industrial Research | 09/860[0137]/2012-EMR-1 | Nimi Marcel |

The funders had no role in study design, data collection and interpretation, or the decision to submit the work for publication.

### Author contributions

NM, Conception and design, Acquisition of data, Analysis and interpretation of data, Drafting or revising the article; AS, Conception and design, Analysis and interpretation of data, Drafting or revising the article

### Author ORCIDs

Apurva Sarin, http://orcid.org/0000-0003-0851-4818

### Ethics

Animal experimentation: All experimental protocols were approved by the Institutional Animal Ethics Committee (NCBS-AEC-AS-6/½012; INS-IAE-2016/01[N]) and are in compliance with the norms of the Committee for the Purpose of Control and Supervision of Experiments on Animals, Govt. of India.

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
