## [Decision Letter]

Thank you for submitting your work entitled "Notch1 regulated autophagy controls survival and suppressor activity of activated T-regulatory cells" for consideration by *eLife*. Your article has been favorably evaluated by Tadatsugu Taniguchi (Senior editor) and three reviewers, one of whom, is a member of our Board of Reviewing Editors.

The reviewers have discussed the reviews with one another and the Reviewing Editor has drafted this decision to help you prepare a revised submission.

Summary:

In this report, the authors demonstrated that Notch signaling regulated mitochondrial organization, survival and suppressor function of activated Treg cells through the activation of autophagy. Furthermore, they confirmed that Notch intracellular domain (NIC) formed a complex with constituents of the autophagy initiation complex and that the overexpression of an effector of autophagy could restore survival and suppressor function in Notch1-deficient Treg cells. Interestingly, they found that Treg specific deletion of Notch1 resulted in development of inflammation because of dysregulation of immune homeostasis.

Essential revisions:

1) They show that Notch pathway and autophagy do not affect Foxp3 expression in activated Treg cells in Figure 5 and Figure 5, respectively. However, recent studies have reported by using genetically modified mice that Notch pathway and autophagy regulate Foxp3 expression in Treg cells (Nat Immunol., 2015, 16:1162-73; Nat Immunol., 2016 AOP). They should discuss this discrepancy including housing condition of mice (How is the detailed housing condition of mice in this manuscript? Are there any infections that affect immune responses?) Furthermore, Treg cells are known to express several signature genes in addition to Foxp3 (e.g. CTLA-4, CD25, Eos, GITR…). Thus, the additional signature genes also need to be tested in Notch1- or autophagy-deficient Treg cells (The global transcriptome analysis would be informative).

2) How are shRNAs transduced into Treg cells in Figure 1? Do the vectors (retroviral vector?) have some fluorescence marker to detect infected cells?

3) In Figure 2, they demonstrate in Figure 2 that Beclin1 formed a complex with Notch1, VPS34 and ATG14 in Treg cells. However, Figure 2 indicates that Notch1 does not bind to VPS34. This discrepancy should be discussed. In addition, cell death of Treg cells was evaluated by culturing Treg cells at 0.3 x 106/ml concentration. This culture condition appears to be relatively sparse in order Treg to interact with Notch ligands expressed on adjacent Treg cells. Are Notch ligands-mediated Notch cleavage involved in the increased cell death in this culture condition? Downregulation of Notch ligands by shRNA would address this point.

4) In Figure 3, they demonstrate that mitochondrial organization is regulated by the Notch-autophagy pathway in Treg cells. However, the contribution of mitochondrial organization to Treg survival and function remains unclear in this study.

5) In Figure 5, they show that Notch1-deficient Treg cells lose their suppressor function in vivo. However, they previously reported that Notch1-deficient Treg cells possessed suppressor activity in in vitro suppression assay (Sci Signal., 2012, 5:ra53). They need to explain this discrepancy between in vitro and in vivo.

6) They have indicated that the overexpression of Atg3 restores cytokine-independent survival of Notch1-deficient Treg cells in Figure 2. Can the ectopic expression of Atg3 rescue the defect of suppressor function in Notch1-deficient Treg cells in Figure 5 and Figure 6?

7) They show that IL-2 withdrawal increases autophagy in Treg cells and evaluated their function in vivo by transferring Tregs with OT-II T cells. Is this in vivo model appropriate for evaluating IL-2 withdrawal state?

8) In Figure 5, which mouse strain was used for isolating Treg cells in co-injection model? Are Treg cells from OT-II mice used? This information is not described in Figure legends and methods sections although the Methods section described that 'activated Treg cells are used'. If authors used in vitro pre-activated Treg cells from B6 mice, did this suppression occur in antigen non-specific manner? As the host mice are not irradiated in Figure 5, the number of two types of donor cells are extremely small compared with host-derived cells. How did the small number of two types of cells interact with each other in vivo, which results in suppression of T cells? It should be explained. The frequency of donor cells relative to host cells should be also shown in Figure 5.

9) Overall, while the data are quite convincing and extremely well-presented in the figures, the text is frequently difficult to read. There are several places where the reader becomes bogged down and has to read a sentence several times before understanding the point the authors are trying to make to their audience. The authors should work hard to rewrite the manuscript using simple declarative sentences whenever possible. Many sentences contain several dependent clauses which makes the interpretation difficult. In other places, even simple sentences are difficult to understand. For example, in the Results section, the sentence states: "We next assess if preventing the activation of autophagy was detrimental to cytokine-independent survival". This sentence is difficult to interpret and perhaps could be more clearly stated as "We asked whether autophagy is important for survival in the absence of cytokines".

10) Since *eLife* is a journal that reaches a wide audience, many readers are not immunologists and the details of experimental systems are likely to be lost on most readers. In the context of this manuscript, the details of the in vivo suppression systems employed in Figure 6 are likely to be confusing to most readers. Simplification of the data described in the subsection “Non-nuclear Notch1 activity is a positive regulator of Treg function” would make these data more accessible to a wider audience.

---

## [Author Response]

1) They show that Notch pathway and autophagy do not affect Foxp3 expression in activated Treg cells in Figure 5 and Figure 5, respectively. However, recent studies have reported by using genetically modified mice that Notch pathway and autophagy regulate Foxp3 expression in Treg cells (Nat Immunol., 2015, 16:1162-73; Nat Immunol., 2016 AOP). They should discuss this discrepancy including housing condition of mice (How is the detailed housing condition of mice in this manuscript? Are there any infections that affect immune responses?) Furthermore, Treg cells are known to express several signature genes in addition to Foxp3 (e.g. CTLA-4, CD25, Eos, GITR…). Thus, the additional signature genes also need to be tested in Notch1- or autophagy-deficient Treg cells (The global transcriptome analysis would be informative).

We have expanded the Discussion section of the revised manuscript to incorporate comments on the studies by Chatila et al. and Wei et al. The Discussion has been extended to include the following points, arising from the comments:

Charbonnier and colleagues (Nat Immunol., 2015, 16:1162-73)position RBPJ-k dependent Notch signaling as a negative regulator of Treg function and implicate non-canonical Notch1 activity in the maintenance of Treg identity. The dynamic regulation of Notch emerging from their experiments, align with our observations and position Notch1 as a critical determinant of Treg differentiation, function and homeostasis. A key difference from this study is that our investigation employed natural Tregs activated in vitro, whereas the Charbonnier group has characterized natural Tregs cells in circulation. The study by Jun Wei et al. (Nat Immunol., 2016, 17:277-286), mapsevents at the earlier stages of Treg activation to define a critical role for autophagy during differentiation of Tregs. Although focusing on different stages of (natural) Treg development, taken together the studies suggest dynamic regulation of Notch activity (Nat Immunol., 2015, 16:1162-73)or autophagy(Nat Immunol., 2016, 17:277-286) for Treg homeostasis. Our experiments, add to these observations by the demonstrated interaction between the two signaling modules in the context of activated Treg survival and function.

Another notable difference between our study and that of Chatila et al. is in the defects in immune function following the ablation of Notch1 in Tregs. We speculate that some of these may arise from the housing conditions of the mice, as our mice are maintained in high barrier but not in SPF1 conditions. We posit that the inflammation observed in Foxp3Notch1^-/-^ mice, results from a breakdown of tolerance and a response to antigens experienced in these conditions. Importantly, genetic (littermate) controls were indistinguishable from wild type mice arguing against overt infection. Interestingly, the phenotypes of autoimmunity reported in Atg7 deficient mice (Nat Immunol., 2016, 17:277-286), are aligned with our observations of lymphoid proliferation and increased numbers of recently activated or memory T-cells in Foxp3Notch1^-/-^ mice. The interaction between Notch and autophagy revealed in our experiments suggests an hitherto unappreciated for Notch signaling in the regulation of this process. Whether this interaction is conserved at all stages of Treg activation remains to be investigated.

Foxp3 stability:

Wei et al.have presented evidence that Atg7 maintains Foxp3 expression in Tregs. While we did not observe a marked difference in Foxp3 expression in the 24-36 hour period following ablation of VpS34, the loss of Foxp3 expression may well be an underlying cause of the deficits observed in suppressor assays.

As reported by Chatila et al., we observe a small but consistent increase in the Treg^+^ subsets in CD4Cre+ Notch1^-/-^ mice (reported in the first paragraph of the subsection “Perturbations of Notch1 or autophagy inhibit suppressor activity in activated Tregs”). Since we do not have evidence of a gain of Treg function, we are unable to comment on the physiological consequences of the differences in this number.

Housing Conditions:

The mouse colonies used in our study are maintained in high barrier – but not SPF conditions – used by Chatila et al. Immune hyperactivity, may be triggered by either low level of antigen experience or activation of autoimmune responses in the conditions we hold our colonies. However, the mice used in our experiments are not infected as cellularity of lymphoid organs and the representation of naïve and memory T-cell subsets, (which reflect immune responses to ongoing infections) are comparable in genetic controls or littermates (Figure 5—figure supplement 1). Hence we conclude that the housing conditions, allows immune responses to innocuous antigens, triggering inflammation in the mice of the affected genotype. Information on housing conditions of mice and the health checks on the colony are described in the Methods (subsection “Mice”) of the revised manuscript.

Furthermore, Treg cells are known to express several signature genes in addition to Foxp3 (e.g. CTLA-4, CD25, Eos, GITR…). Thus, the additional signature genes also need to be tested in Notch1- or autophagy-deficient Treg cells (The global transcriptome analysis would be informative).

A global gene expression analysis of Tregs from Notch deficient and control mice has been initiated and is currently ongoing. The analysis included activated Tregs of the two genotypes, where the ablation of Notch1 was under the control of the CD4 promoter. In light of the aforementioned studies, we have expanded the scope of the analysis to examine freshly isolated cells as well as cells deficient in Notch1 under the control of the Foxp3 promoter. Mice, in the latter group is generated in very low numbers in our hands and hence the completion of the analysis is pending availability of enough mice of this genotype.

Appreciating the importance of this analysis, the revised manuscript now includes a comparison of 15 Treg subset specific genes, confirmed by RT-PCR analysis or flow cytometry for proteins in Figure 5 in the revised manuscript. The corresponding data from the gene expression analysis of these are shown Figure 5—figure supplement 1. While we do not observe differences, we are unable to comment definitively unless the more detailed analysis of the groups mentioned is completed.

2) How are shRNAs transduced into Treg cells in Figure 1? Do the vectors (retroviral vector?) have some fluorescence marker to detect infected cells?

The detailed protocol is now included in the section in Methods (subsection “T-cells and retroviral transductions”).Infected cells are selected by antibiotic selection using puromycin resistance.

3) In Figure 2, they demonstrate in Figure 2 that Beclin1 formed a complex with Notch1, VPS34 and ATG14 in Treg cells. However, Figure 2 indicates that Notch1 does not bind to VPS34. This discrepancy should be discussed.

This comment highlights a point that was not adequately addressed in the first submission. Vps34 and Beclin are known to form multiple and dynamic cellular complexes, which are regulated by the phosphorylation of the proteins themselves as well their interacting partners (Kang et al., 2011; Kim et al., 2012). Thus the exclusion of Vps34 from the NIC complex reveals specific interactions with proteins (such as Beclin) whose functions may be regulated by Notch activity. The data also do not rule out transient associations between NIC and Vps34. The data support the evidence of Notch interactions with autophagy intermediates, but the functional significance of specific or dynamic complexes that include Notch remains to be investigated further.

The text has been modified in the fourth paragraph of the subsection “Notch1 activity regulates autophagy in Tregs”.

In addition, cell death of Treg cells was evaluated by culturing Treg cells at 0.3 x 106/ml concentration. This culture condition appears to be relatively sparse in order Treg to interact with Notch ligands expressed on adjacent Treg cells. Are Notch ligands-mediated Notch cleavage involved in the increased cell death in this culture condition? Downregulation of Notch ligands by shRNA would address this point.

The concentration at which cells are cultured for the deprivation assay in 48 well plates or in 24 well plates permits cell contact as this can be confirmed by visual analysis. The requirement for inputs from the DLL-1 ligand in Treg survival following growth factor withdrawal was identified using this format and published earlier (Perumalsamy et al., 2012), hence the data are not included in this manuscript. The requirement for DLL1 in the regulation of mitochondrial integrity (shown in Figure 3) was tested based on these earlier observations.

4) In Figure 3, they demonstrate that mitochondrial organization is regulated by the Notch-autophagy pathway in Treg cells. However, the contribution of mitochondrial organization to Treg survival and function remains unclear in this study.

In the revised submission, we include evidence that acute disruption of Notch signaling by the addition of the gamme-secretase inhibitor (GSI-X), triggers a loss in mitochondrial trans-membrane potential as early at 8 hours following cytokine withdrawal. The new data are shown in Figure 3 in the manuscript, and provide evidence that Notch activity regulates mitochondrial health. The addition of GSI eventually results in cell death as shown in Figure 2.

As we are yet to identify the mechanism by which Notch activity signals to mitochondria we are currently unable to demonstrate a causal relationship between mitochondrial integrity (function and organization) and Treg suppressor function. However, a link is supported by the evidence that recombinant non-nuclear NIC or Atg3, restore both mitochondrial integrity (Figure 3) and suppressor activity (new Figure 5 and Figure 6) in Notch deficient Tregs. We project a role for Notch in maintenance of mitochondrial integrity, based on these data and the current appreciation of the role of mitochondria in cell survival.

5) In Figure 5, they show that Notch1-deficient Treg cells lose their suppressor function in vivo. However, they previously reported that Notch1-deficient Treg cells possessed suppressor activity in in vitro suppression assay (Sci Signal., 2012, 5:ra53). They need to explain this discrepancy between in vitro and in vivo.

Based on the understanding of Notch activity in Tregs we speculate that Notch deficient Tregs may be protected from cell death in the suppressor assay performed in the culture dish as levels of cytokines and growth factors can build up in the dish. As Foxp3 levels are not compromised in Notch^-/-^Tregs, this will allow suppressor activity.

6) They have indicated that the overexpression of Atg3 restores cytokine-independent survival of Notch1-deficient Treg cells in Figure 2. Can the ectopic expression of Atg3 rescue the defect of suppressor function in Notch1-deficient Treg cells in Figure 5 and Figure 6?

The experiments with Atg3 are included in the new Figure 5 and Figure 6—figure supplement 1 in the revised manuscript. In both systems, Atg3 restores the defect in suppressor function.

7) They show that IL-2 withdrawal increases autophagy in Treg cells and evaluated their function in vivo by transferring Tregs with OT-II T cells. Is this in vivo model appropriate for evaluating IL-2 withdrawal state?

The analysis of IL-2 withdrawal in vitro is a model to examine cell intrinsic processes that regulate survival and homeostasis in the T-cell lineage. This assay had uncovered unusual signaling by Notch in the Treg subset (Perumalsamy et al., 2012) and revealed the interaction with autophagy as reported in the current study.

Cell survival in complex and changing environments associated with inflammation is an important component determining cell function. The identification of the Notch-Autophagy axis for cell survival motivated the experiments to assess this interaction in the contexts of suppressor function. We capitalized on the ability to manipulate these molecules in isolated cells, and test consequences for function in different models. Our conclusions are strengthened by the recent observations using other approaches (Wei et al., Nat Immunol., 2016, 17:277-286I), ascribing a critical requirement for autophagy in Treg homeostasis.

8) In Figure 5, which mouse strain was used for isolating Treg cells in co-injection model? Are Treg cells from OT-II mice used? This information is not described in Figure legends and methods sections although the Methods section described that 'activated Treg cells are used'. If authors used in vitro pre-activated Treg cells from B6 mice, did this suppression occur in antigen non-specific manner? As the host mice are not irradiated in Figure 5, the number of two types of donor cells are extremely small compared with host-derived cells. How did the small number of two types of cells interact with each other in vivo, which results in suppression of T cells? It should be explained. The frequency of donor cells relative to host cells should be also shown in Figure 5.

We apologize that this information was not presented in a concise manner in the original submission. The information on the source of mice and additional experimental detail are included in the Methods and legends. The supporting text has also been modified for clarity.

The Tregs are not derived from the OT-II mice and hence to that extent suppression is not determined by the TCR specificity of the Tregs and responder OT-II cells. However, multiple modes of suppression are associated with Tregs and with dendritic cells or CD4 or CD8 T-cells (which is not antigen-specific) as well as modifying the cytokine milieu are thought to be some of the mechanisms by which Tregs mediate suppression. The protocol for the suppressor assay we have used is as established by others (Quah et.al, 2007; Klein et al., 2003).

In addition, suppressor activity has also been validated in another model using the IPGTT assay.

The revised submission includes data to show that the two cell types – OT-II and Tregs – are detected in the same host lymph nodes (Figure 5—figure supplement 1). We have used molecular markers to track cells using flowcytometry, which allows for the detection of relatively low numbers of cells as can be seen in the plot in the supplementary figure. In the experiments contributing data in the study, we do not collect information on Tregs but use a combination of a cell surface CD45.2 marker and CFSE fluorescence to track OT-II cells thereby excluding the host and transferred Tregs from the analysis, respectively.

9) Overall, while the data are quite convincing and extremely well-presented in the figures, the text is frequently difficult to read. There are several places where the reader becomes bogged down and has to read a sentence several times before understanding the point the authors are trying to make to their audience. The authors should work hard to rewrite the manuscript using simple declarative sentences whenever possible. Many sentences contain several dependent clauses which makes the interpretation difficult. In other places, even simple sentences are difficult to understand. For example, in the Results section, the sentence states: "We next assess if preventing the activation of autophagy was detrimental to cytokine-independent survival". This sentence is difficult to interpret and perhaps could be more clearly stated as "We asked whether autophagy is important for survival in the absence of cytokines".

We apologize for the difficulty in accessing information presented in the manuscript. The text has been revised bearing in mind this comment and we hope the changes introduced will address this concern to a large extent.

10) Since eLife is a journal that reaches a wide audience, many readers are not immunologists and the details of experimental systems are likely to be lost on most readers. In the context of this manuscript, the details of the in vivo suppression systems employed in Figure 6 are likely to be confusing to most readers. Simplification of the data described in the subsection “Non-nuclear Notch1 activity is a positive regulator of Treg function” would make these data more accessible to a wider audience.

The text in this section has been re-written and an introduction to the suppressor assay included in this section as well as statements clarifying the underlying motivation of the different conditions included in the analysis. We trust these will substantially address the concerns raised.